# Structural basis of *Plasmodium vivax* inhibition by antibodies binding to the circumsporozoite protein repeats

Iga Kucharska[1], Lamia Hossain[1,2], Danton Ivanochko[1], Qiren Yang[1], John L Rubinstein[1,2,3], Régis Pomès[1,2], Jean-Philippe Julien[1,2,4]*

[1]Program in Molecular Medicine, The Hospital for Sick Children Research Institute, Toronto, Canada; [2]Department of Biochemistry, University of Toronto, Toronto, Canada; [3]Department of Medical Biophysics, University of Toronto, Toronto, Canada; [4]Department of Immunology, University of Toronto, Toronto, Canada

**Abstract** Malaria is a global health burden, with *Plasmodium falciparum* (Pf) and *Plasmodium vivax* (Pv) responsible for the majority of infections worldwide. Circumsporozoite protein (CSP) is the most abundant protein on the surface of *Plasmodium* sporozoites, and antibodies targeting the central repeat region of CSP can prevent parasite infection. Although much has been uncovered about the molecular basis of antibody recognition of the PfCSP repeats, data remains scarce for PvCSP. Here, we performed molecular dynamics simulations for peptides comprising the PvCSP repeats from strains VK210 and VK247 to reveal how the PvCSP central repeats are highly disordered, with minor propensities to adopt turn conformations. Next, we solved eight crystal structures to unveil the interactions of two inhibitory monoclonal antibodies (mAbs), 2F2 and 2E10.E9, with PvCSP repeats. Both antibodies can accommodate subtle sequence variances in the repeat motifs and recognize largely coiled peptide conformations that also contain isolated turns. Our structural studies uncover various degrees of Fab-Fab homotypic interactions upon recognition of the PvCSP central repeats by these two inhibitory mAbs, similar to potent mAbs against PfCSP. These findings augment our understanding of host-*Plasmodium* interactions and contribute molecular details of Pv inhibition by mAbs to unlock structure-based engineering of PvCSP-based vaccines.

*For correspondence:
jean-philippe.julien@sickkids.ca

**Competing interest:** The authors declare that no competing interests exist.

## Editor's evaluation

This paper combines simulations and experimental biophysical approaches to probe the molecular basis of antibody recognition of key repeat peptides found on the surface of *Plasmodium vivax*, a parasite that causes malaria. Currently, we know surprisingly little about the mechanism by which antibodies recognize these peptides. Thus, this work provides crucial molecular insights into a recognition process of considerable biomedical significance, which might ultimately inform rational design of novel and more effective antimalarial vaccines.

## Introduction

Malaria is a major public health concern, with an estimated 409,000 deaths in 2019 (***World Health Organization, 2020***). Human malaria is caused by *Plasmodium* parasites, with the majority of cases attributed to *Plasmodium falciparum* (Pf) and *Plasmodium vivax* (Pv) (***Lover et al., 2018***). Pv is the predominant *Plasmodium* spp. in circulation for a majority of countries outside of Africa (~75% of cases in South and North America, ~50% of cases in the Southeast Asia region, and ~30% in the Eastern Mediterranean region; ***World Health Organization, 2020***). Despite overall lower mortality

**Figure 1.** Comparison of PvCSPvk210 and PvCSPvk247 repeat sequences. (**A**) Schematic representations of PvCSPvk210 and PvCSPvk247 sequences, each including an N-terminal domain, central repeat region, and C-terminal domain. Colored blocks represent repeat motifs. The sequence of peptides used in circular dichroism (CD) spectroscopy studies is shown below. (**B**) Superimposition of the CD spectra obtained for PvCSPvk210 (top panel) and PvCSP247 peptides (bottom panel). PvCSPvk210 peptides 210-1, 210-2, 210-3, 210-4, and 210-5 are colored from navy to light blue; PvCSPvk247 peptides are depicted in green (247-1), yellow (247-2), and orange (247-3), respectively.

compared to Pf malaria, Pv infection can cause debilitating disease, including fever, myalgia, chronic anemia, reduced birthweight, and increased risk of neonatal death (*Alexandre et al., 2010*; *Bardají et al., 2017*; *Genton et al., 2008*).

Circumsporozoite protein (CSP) is the most abundant protein on the surface of all *Plasmodium* sporozoites and is necessary for parasite development and infection (*Cerami et al., 1992*; *Ménard et al., 1997*; *Nguitragool et al., 2017*). CSP contains an unusual central region consisting of multiple, short amino acid repeats whose sequence depends on the *Plasmodium* species (*Chenet et al., 2012*; *Rich et al., 2000*; *Tahar et al., 1998*). The PvCSP central region is composed of nonapeptides GDRA(A/D)GQPA and ANGAGNQPG characteristic of strains VK210 and VK247, respectively (*Arnot et al., 1985*; *Rosenberg et al., 1989*; *Figure 1A*). Unlike the 4-amino acid (aa)-long motifs of Pf and *Plasmodium berghei* (Pb) CSP, which are rich in asparagine and proline residues, PvCSP repeats are longer and consist primarily of glycine and alanine residues (~50% of all residues). Moreover, ~26% of the PvCSPvk210 central repeat region consists of charged residues, including arginine and aspartic acid residues. Both VK210 and VK247 strains have worldwide distribution (*Cheng et al., 2013*; *Kain et al., 1992*; *Soares et al., 2020*), and VK210 appears to be a major target of the humoral immune response in studied populations (*González et al., 2001*; *Kim et al., 2010*; *Soares et al., 2020*).

The central region of CSP is immunodominant, and antibodies targeting the repeats can inhibit sporozoite infection (*Imkeller et al., 2018*; *Kisalu et al., 2018*; *Kisalu et al., 2018*; *Mishra et al., 2012*; *Murugan et al., 2020*; *Oyen et al., 2017*; *Oyen et al., 2020*; *Pholcharee et al., 2020*; *Pholcharee et al., 2021*; *Potocnjak et al., 1980*; *Tan et al., 2018*; *Triller et al., 2017*; *Wang et al., 2020*). The PfCSP central repeat is a major component of the most advanced malaria vaccine to date, RTS,S/AS01 (*Adepoju, 2019*; *Draper et al., 2018*), and PvCSP or PvCSP-derived peptides are present in various clinical vaccine candidates against Pv (*Mueller et al., 2009*; *Mueller et al., 2015*). Effective pre-erythrocytic stage vaccines are highly desirable to reduce the number of primary infections, and

in the case of Pv, to consequently prevent the establishment of hypnozoites that can lead to multiple relapses (*Krotoski et al., 1982*).

Although not much is known about the molecular basis of PvCSP recognition by inhibitory antibodies, a positive statistical association between the level of antibodies against the repeat region of PvCSP and protection has previously been established (*Yadava et al., 2014*). Anti-PvCSP species-specific monoclonal antibodies (mAbs) 2F2 and 2E10.E9 were generated in mice after immunization with radiation-attenuated Pv sporozoites of strains VK210 and VK247, respectively (*Nardin et al., 1982*). Incubation of Pv sporozoites with mAbs 2F2 and 2E10.E9 results in significantly reduced sporozoite infectivity, and thus despite a limited molecular understanding of their recognition, both antibodies have been valuable research tools in studies of Pv sporozoites (*Cabrera-Mora et al., 2015*; *Gimenez et al., 2017*; *Miyazaki et al., 2020*; *Roth et al., 2018*; *Teixeira et al., 2014*). Interestingly, a recent study reported that sporozoites attenuated with low concentrations of mAb 2F2 were significantly reduced in size and had lower DNA content, indicating post-hepatocyte-invasion antibody inhibition of liver stage development (*Roth et al., 2018*).

Here, we present a detailed molecular analysis of the PvCSP repeat region and its recognition by inhibitory mAbs 2F2 and 2E10.E9. Molecular dynamics (MD) simulations on PvCSP-derived repeat peptides indicate that in the absence of interacting mAbs, the PvCSP repeat is largely disordered. Our structural studies reveal how mAbs 2F2 and 2E10.E9 lock PvCSP repeat peptides in a predominant coiled conformation, with antibody germline-encoded aromatic residues contributing significantly to the antigen contacts. Moreover, we describe how mAb 2E10.E9 engages in head-to-head homotypic interactions when targeting PvCSP in a similar manner as previously described human mAbs against PfCSP (*Imkeller et al., 2018*; *Oyen et al., 2018*; *Pholcharee et al., 2021*) and a murine mAb against PbCSP (*Kucharska et al., 2020*).

## Results

### PvCSP repeat peptides are structurally disordered and behave like harmonic springs

Due to the variety of PvCSP repeat sequence motifs, we created an extensive list of peptides for circular dichroism (CD) spectroscopy and MD simulations studies (*Supplementary file 1*, *Figure 1A*) to examine the structural propensities of 18- and 27-aa-long motifs from the PvCSPvk210 and PvCSPvk247 repeats. CD spectra of all analyzed peptides were indicative of a lack of secondary structure, with minima at ~200 nm (*Figure 1B*).

For a finer dissection of potential minor secondary structure propensities, we performed all-atom MD simulations on 27-aa peptides derived from PvCSPvk210 and PvCSPvk247 (*Supplementary file 1*, *Figure 2*, *Figure 2—figure supplement 1*, *Figure 2—figure supplement 2*). All analyzed peptides were highly disordered in solution and adopted a large ensemble of conformations (*Figure 2A*), similar to peptides derived from PfCSP and PbCSP (*Kucharska et al., 2020*). Secondary structure was present mainly in the form of transient hydrogen-bonded β-turns (*Figure 2B*). The propensity of individual residues to form turns varied from 0 to ~60%, with PvCSPvk210 peptides displaying slightly lower averaged turn propensity (~15–20%) than PvCSPvk247 peptides (~23–26%) (*Figure 2C*). PvCSPvk210 peptides containing (GDRAAGQPA)$_2$ motifs (210-7 and -9), as well as the last repeat of the 247-3 peptide (GNGAGGQAA), were the only motifs with a significant propensity to form helices (>20%).

In order to estimate the elastic modulus of these peptides with intrinsically low secondary structure propensities, peptides were modeled as Hookean springs (*Figure 2D*), in which the force needed to extend or compress a spring by some distance is proportional to that distance, and the underlying energy function is quadratic (*Equation 1*). The PMF or free energy governing changes in the peptide's end-to-end distance was computed and fit to a quadratic function. The excellent data fits ($R^2$ = 0.91–0.97) suggest that in aqueous solution, peptides derived from the PvCSP repeats behave like harmonic springs. On average, the elastic modulus was ~3–5 cal/(mol Å$^2$), with the highest values (stiffest peptide) observed for peptide 247-2 (4.9 ± 0.2 cal/(mol Å$^2$)) and lowest (most flexible peptide) observed for peptide 247-1 (3.2 ± 0.2 cal/(mol Å$^2$)).

### Molecular basis of 2F2 recognition of the PvCSPvk210 central repeat

To gain insights into how inhibitory mAb 2F2 binds to a largely disordered, elastic repeating PvCSPvk210 epitope, we performed isothermal titration calorimetry (ITC) with PvCSPvk210 repeat peptides

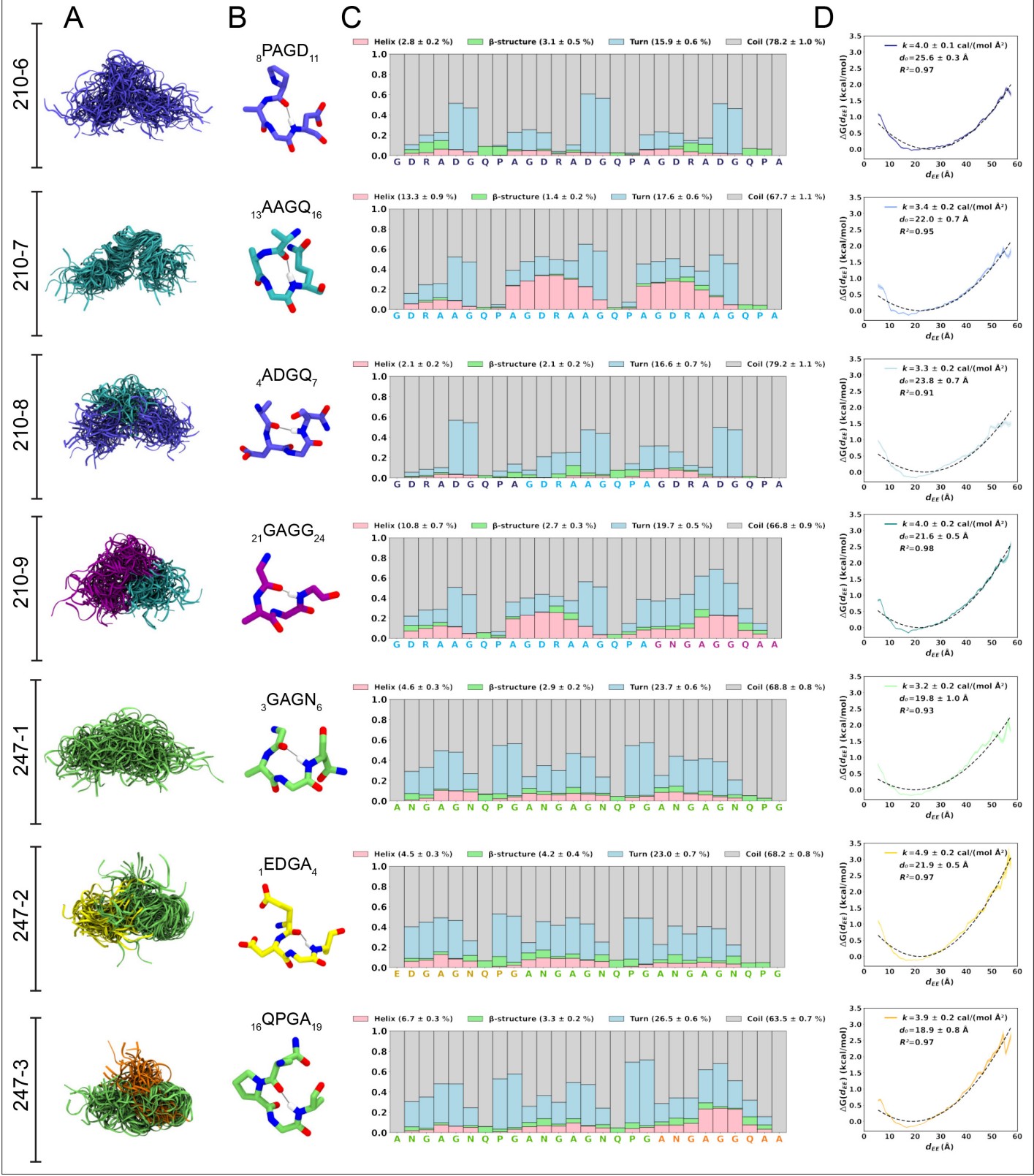

**Figure 2.** Conformational flexibility of PvCSPvk210 and PvCSPvk247 peptides. (**A**) Superposition of the conformations of peptides resulting from molecular dynamics (MD) simulations at every 2 ns and aligned to the conformational median structure. (**B**) Example snapshots of the highest (peptides 210-7, 210-8, 210-9, and 247-3) or the second highest (peptides 210-6, 247-1, 247-2) propensity β-turn for each peptide. Color coding of atoms: oxygen (red), nitrogen (blue), and hydrogen (white). For clarity, only the hydrogen atom involved in the H-bonded turn is shown. H-bond between C=O of

*Figure 2 continued on next page*

*Figure 2 continued*

residue $i$ and N–H of residue $i + 3$ is shown as a gray line. (**C**) Secondary structure propensity at each residue, averaged over 20 replicas and computed using the Dictionary of Secondary Structure for Proteins (DSSP) criteria (**Nagy and Oostenbrink, 2014**). (**D**) Elastic modulus of peptides computed from MD simulations. The reversible work or free energy (ΔG) for extension and compression of the peptide is plotted as a function of equilibrium end-to-end distances ($d_{EE}$) (solid line). The data is fitted to the quadratic function of elastic potential energy (black dashed line). For each peptide, the estimated values of $k$ (elastic modulus), $d_0$ (equilibrium $d_{EE}$), and $R^2$ (regression coefficient to indicate quality of fit) are shown. Shading represents standard error of mean.

The online version of this article includes the following figure supplement(s) for figure 2:

**Figure supplement 1.** Time evolution of structural properties of PvCSP peptides.

**Figure supplement 2.** Ensemble-averaged backbone-backbone H-bonding maps for each PvCSP peptide sequence.

representative of the different sequence motifs contained within the central repeat region (**Supplementary file 1**, **Figure 3A**, **Figure 3—figure supplement 1**). 2F2 Fab binds to all tested peptides containing two repeats with similar affinity, with a slight preference for peptide 210-3 (GDRADGQ-PAGDRAAGQPA; 13.8 nM) over 210-2 (GDRAAGQPAGDRAAGQPA; 37.5 nM) (**Figure 3A**), with other peptides displaying intermediate affinities (17.1–19.1 nM). Peptides 210-3 and 210-2 differ in the fifth residue of the first repeat, which is an aspartic acid for peptide 210-3 and an alanine for peptide 210-2 (GDRA(D/A)GQPA).

Next, we co-crystalized the 2F2 Fab with five different peptides derived from PvCSPvk210 (210-1, 210-2, 210-3, 210-4, and 210-5, **Supplementary file 1**, **Figure 3**, **Figure 3—figure supplement 2**) to gain molecular insights into the binding mode and cross-reactivity of mAb 2F2 binding to PvCSPvk210. The crystal structures were solved to resolutions ranging from 1.97 Å to 2.67 Å (**Table 1**). 2F2 recognizes the core epitope ($_2$DRA(D/A)GQ$\underline{PAGD}_{11}$) of all PvCSPvk210 peptides in an almost identical coil conformation peptide backbone root-mean-square deviation (RMSD 0.10–0.28 Å) with two consecutive β-turns observed for residues $_7$QPAGD$_{11}$ (**Supplementary file 2**) forming one turn of a $3_{10}$-helix, thus consistent with the moderate secondary structure propensities observed for the unliganded PvCSPvk210 repeats (**Figure 3B**). The co-crystal structures also provide molecular insights into the cross-reactivity of 2F2 to different types of PvCSPvk210 repeat motifs ($_1$GDRA(D/A)GQPA$_9$). The sidechains of D/A$_5$ point up and away from the 2F2 paratope and do not significantly contribute to the 2F2 Fab-peptide interaction, helping to explain the similar binding affinities to the different peptides containing this variation (**Figure 3E**). 2F2 also binds the 210-5 peptide containing a unique repeat C-terminal of the central region ($_1$GDRAAGQPAG$\underline{NGA}$G-$\underline{GQA}$A$_{18}$); however, the electron density of residues 12–18 C-terminal of the bound core peptide motif is weak in the co-crystal structure, thus providing limited structural insight into this peptide region and suggesting it does not make strong interactions with 2F2 (**Figure 3—figure supplement 2**).

The recognition of PvCSPvk210 peptides by 2F2 is mediated mostly by residues localized in heavy chain complementarity-determining regions (HCDRs) 1, 2, and 3, and kappa chain complementarity-determining regions (KCDRs) 1 and 3. The PvCSPvk210 one turn of a $3_{10}$-helix is positioned in the hydrophobic pocket formed by KCDR1 residues of the antibody (**Figure 3C and D**, **Figure 3—figure supplement 3C, E, and G**), and is stabilized by three H-bonds formed between A$_9$ and the backbone of K.Gly91 and K.Phe96, and G$_{10}$ and the sidechain of H.Ser58 (**Figure 3E**). The antibody-antigen complex buries 848 Å² on the Fab (437 Å² on HC and 411 Å² on KC) and 1023 Å² on the 210-4 peptide. Peptide residues $_1$GRDADG$_6$ are positioned between HCDR1 and 3 and do not interact with the light chain (**Figure 3C and D**).

2F2 binds to PvCSPvk210 peptides using both germline and somatically hypermutated residues. Nine germline-encoded aromatic residues form significant van der Waals interactions with the peptides, contributing a total of ~385 Å² of buried surface area (BSA; **Figure 3E**, **Figure 3—figure supplement 3C, E, and G**). To accommodate arginine residues present in the sequence of the PvCSPvk210 peptides, the 2F2 paratope has an overall electronegative potential (**Figure 3—figure supplement 4A**). R$_3$ and Q$_7$ play a central role in mediating the Fab-peptide interactions, forming six H-bonds with both heavy and light chain residues of the antibody (**Figure 3E**), and contributing ~107 Å² and ~ 156 Å² of BSA, respectively.

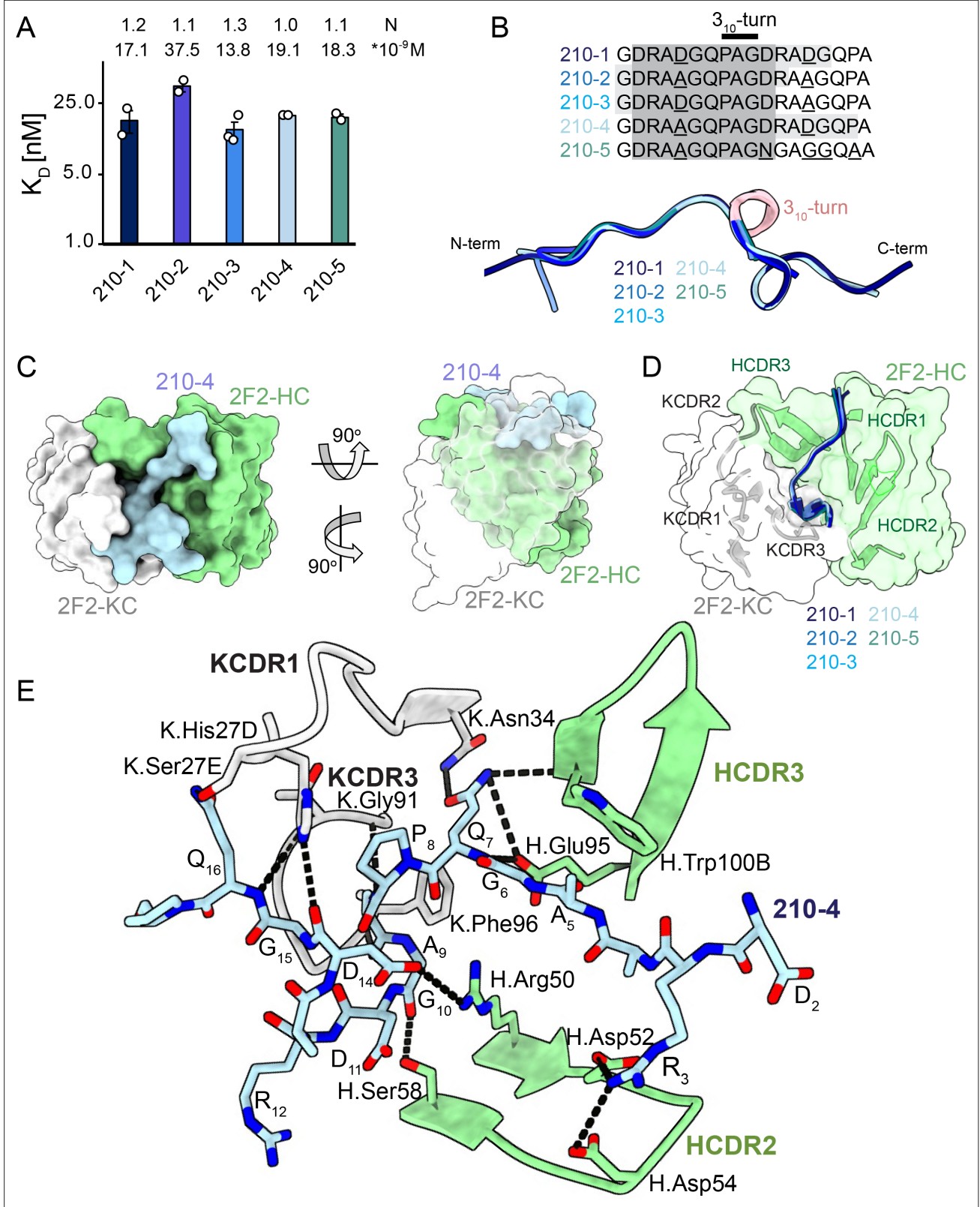

**Figure 3.** 2F2 Fab binding to PvCSPvk210 repeat peptides. (**A**) Affinities of 2F2 Fab for peptides 210-1, 210-2, 210-3, 210-4, and 210-5 as measured by isothermal titration calorimetry (ITC). Open circles represent independent measurements. Mean binding constant ($K_D$) and binding stoichiometry (N) values are shown above the corresponding bar. Error bars represent SEM. (**B**) Upper panel: sequences of peptides used in ITC experiments with variable residues underlined. Dark gray denotes the core epitope of the peptide resolved in all the X-ray crystal structures, and light gray shading

*Figure 3 continued*

indicates residues resolved in the corresponding X-ray crystal structures. Bottom panel: comparison of the conformations of PvCSP210 peptides in X-ray crystal structures. PvCSPvk210 peptides are colored from navy to light blue, with the residues adopting one turn of a $3_{10}$-helix depicted in pink. (**C**) Top and side views of the 210-4 peptide (light blue) in the binding groove of the 2F2 Fab shown as surface representation (heavy chain [HC] shown in green and kappa chain [KC] shown in white). (**D**) Comparison of the conformations adopted by the core epitope of peptides 210-1, 210-2, 210-3, 210-4, and 210-5 when bound to 2F2. (**E**) Detailed interactions between Fab 2F2 and peptide 210-4. H-bonds and salt bridges are shown as black dashes, peptide 210-4 is shown in light blue, HC is shown in green, and KC is shown in gray. Fab residues are annotated with H or K letters to indicate heavy and kappa light chain, respectively.

The online version of this article includes the following figure supplement(s) for figure 3:

**Figure supplement 1.** Isothermal titration calorimetry (ITC) measurements of 2F2 Fab binding to peptides 210-1 (**A**), 210-2 (**B**), 210-3 (**C**), 210-4 (**D**), and 210-5 (**E**).

**Figure supplement 2.** Stereo-image of the composite omit map electron density contoured at 1.0–1.2 sigma for peptides 210-1 (**A**), 210-2 (**B**), 210-3 (**C**), 210-4 (**D**), and 210-5 (**E**).

**Figure supplement 3.** Interactions between germline-encoded aromatic residues in anti-CSP monoclonal antibodies (mAbs) and repeat peptides.

**Figure supplement 4.** Electrostatic surface potential of monoclonal antibodies (mAbs) 2F2 (**A**), 2E10.E9 (**B**), 1210 (*Imkeller et al., 2018*) (**C**), and 3D11 (*Kucharska et al., 2020*) (**D**) bound to CSP peptides.

## Molecular basis of mAb 2E10.E9 recognition of the PvCSPvk247 central repeat

Unlike PvCSPvk210, PvCSPvk247 contains one predominant repeat motif; 'ANGAGNQPG,' with the exception of the first (EDGAGNQPG) and the last repeat (ANGAGGQAA) in the central region (*Figure 1A*). 2E10.E9 Fab binds peptides derived from PvCSPvk247 with affinity ranging from 0.507 µM (peptide 247-4) to 1.67 µM (peptide 247-1), as measured by ITC. Based on the binding stoichiometry derived from ITC, the 18-aa peptide 247-4 accommodates binding of one Fab, while 27-aa peptides 247-1, 247-2, and 247-3 are long enough to allow two Fabs to bind simultaneously (*Figure 4A*, *Figure 4—figure supplement 1*).

We obtained co-crystal structures of the 2E10.E9 Fab in complex with three different peptides: 247-3 and 247-4 at ~2.7 Å resolution, and 247-2 at 3.19 Å resolution (*Table 1*, *Figure 4*, *Figure 3— figure supplement 3B, D, and F*, *Figure 4—figure supplement 2*). In the 2E10.E9 Fab-247-2 co-crystal structure, two Fabs bind to one peptide, which is in agreement with the 2:1 stoichiometry established by ITC (*Figure 4A*). The core epitope of PvCSPvk247 peptides contains eight residues ($_3$GAGNQPGA$_{10}$) and adopts a similar coil conformation in all analyzed peptides when bound to 2E10.E9 (peptide backbone RMSD 0.48–1.0 Å), with only isolated turns (*Figure 4B*, *Supplementary file 2*). Although 2E10.E9 Fab binds peptides 247-2 and 247-3 containing the first (<u>ED</u>GAGNQPG) and the last repeat (ANGAG<u>G</u>Q<u>AA</u>) motifs, the electron density of residues unique to these repeats was absent in the co-crystal structures obtained. This suggests that the antibody does not interact extensively with these variable residues that are outside the well-resolved conserved core (*Figure 4B*, *Figure 4—figure supplement 2*).

2E10.E9 interacts with the PvCSPvk247 peptides using HCDRs 1, 2, and 3, and KCDR1 and 3, with the N-terminal part of the peptide positioned between HCDR1 and 3. Six germline-encoded aromatic residues, including K.Tyr32, K.Tyr92, K.Tyr94, K.Phe96, H.Tyr32, and H.Trp50, play a central role in peptide recognition, forming three H-bonds with $G_9$, Ala$_{10}$, and A$_{13}$, and contributing 176 Å$^2$ BSA to the relatively small paratope of this Fab (444 Å$^2$ total BSA on the 2E10.E9 paratope; 588 Å$^2$ total BSA on peptide 247-3) (*Figure 3—figure supplement 3B, D, and F*). Interestingly, HCDR3 H.Cys98 and H.Cys100 make a disulfide bond that positions H.Gly99 in an ideal position to form an H-bond with residue $G_9$ of the peptide and mimic the stacking effect provided by aromatic sidechains (*Figure 4E*). Residue $N_6$ of the PvCSPvk247 peptides is central to the interaction, forming four H-bonds with HCDR2 residues H.Thr30, H.Asn52, and H.Ser52A (*Figure 4E*).

## Homotypic 2E10.E9 Fab-Fab interactions upon PvCSPvk247 repeat binding

In the 2E10.E9 Fab-247-2 peptide co-crystal structure where two Fabs bind to one peptide, we observed multiple contacts between the two 2E10.E9 Fabs (*Figure 5*). Indeed, the two 2E10.E9 Fabs interact in a head-to-head binding mode at an ~146° angle (*Figure 5A*). Contacts between the two

**Table 1.** X-ray crystallography data collection and refinement statistics.

| | 2F2-210-1 | 2F2-210-2 | 2F2-210-3 | 2F2-210-4 | 2F2-210-5 | 2E10-247-2 | 2E10-247-3 | 2E10-247-4 |
|---|---|---|---|---|---|---|---|---|
| Beamline | APS 23-ID-B | APS 23-ID-D | APS 23-ID-B | APS 23-ID-B | APS 23-ID-D | APS 23-ID-B | APS 23-ID-B | APS 23-ID-D |
| Wavelength (Å) | 1.033167 | 1.033167 | 1.033167 | 1.033167 | 1.033167 | 1.033167 | 1.033167 | 1.033200 |
| Space group | P 1 | C 2 | C 2 | P 1 | C 2 | $P\ 3_1$ | $P\ 2_1$ | P 1 |
| Cell dimensions | 71.5, 81.4, 82.3 | 92.9, 60.4, 158.3 | 92.6, 60.8, 81.4 | 71.7, 82.3, 82.8 | 93.4, 60.5, 159.1 | 142.4, 142.4, 91.3 | 56.4, 144.4, 60.5 | 54.5, 66.3, 142.3 |
| α, β, γ (°) | 94.6, 114.1, 111.6 | 90, 101.5, 90 | 90, 101.6, 90 | 95.3, 113.8, 111.5 | 90, 101.2, 90 | 90, 90, 120 | 90, 102.8, 90 | 100.4, 92.3, 91.7 |
| Resolution (Å)* | 29.48–2.20 (2.25–2.20) | 29.21–2.54 (2.65–2.54) | 29.71–1.97 (2.02–1.97) | 29.61–2.67 (2.77–2.67) | 29.69–2.27 (2.34–2.27) | 29.55–3.19 (3.30–3.19) | 29.48–2.68 (2.78–2.68) | 29.34–2.71 (2.81–2.71) |
| No. molecules in the asymmetric unit (ASU) | 3 | 2 | 1 | 3 | 2 | 2 | 2 | 4 |
| No. observations | 264,344 (16,542) | 181,285 (21,112) | 101,891 (4384) | 155,413 (16,123) | 471,275 (38,772) | 693,999 (70,530) | 184,740 (18,748) | 552,431 (55,667) |
| No. unique observations | 75,088 (4440) | 28,642 (3484) | 31,038 (1962) | 43,179 (4542) | 40,512 (3722) | 34,411 (3469) | 26,439 (2627) | 53,126 (5316) |
| Multiplicity | 3.5 (3.7) | 6.3 (6.1) | 3.3 (2.2) | 4.7 (1.5) | 11.6 (10.3) | 20.1 (20.3) | 7.0 (7.1) | 10.4 (10.5) |
| $R_{merge}$ (%)† | 15.5 (95.2) | 16.4 (144.5) | 6.5 (55.7) | 13.9 (62.8) | 34.2 (175.0) | 36.1 (370.6) | 21.6 (158.6) | 22.4 (123.5) |
| $R_{pim}$ (%) ‡ | 7.5 (32.4) | 10.6 (97.6) | 6.1 (46.9) | 7.8 (30.8) | 15.3 (83.0) | 8.2 (84.0) | 8.8 (63.7) | 7.3 (39.8) |
| $<I/\sigma\ I>$ | 5.2 (1.5) | 6.7 (1.5) | 8.3 (1.5) | 4.7 (1.5) | 9.4 (1.5) | 13.2 (1.7) | 9.2 (1.6) | 7.4 (1.6) |
| $CC_{1/2}$ | 0.965 (0.527) | 0.994 (0.529) | 0.996 (0.603) | 0.987 (0.669) | 0.927 (0.435) | 0.998 (0.756) | 0.993 (0.578) | 0.995 (0.729) |
| Completeness (%) | 97.5 (97.1) | 99.8 (100.0) | 98.7 (90.5) | 98.3 (97.8) | 99.9 (100.0) | 99.8 (99.6) | 99.9 (100.0) | 100.0 (100.0) |
| *Refinement statistics* | | | | | | | | |
| Reflections used in refinement | 74,672 | 28,604 | 31,030 | 43,160 | 40,501 | 34,411 | 26,439 | 53,126 |
| Reflections used for R-free | 3731 | 1432 | 1552 | 2162 | 2026 | 1732 | 1314 | 2067 |
| Non-hydrogen atoms | 10,580 | 6872 | 3595 | 10,504 | 6941 | 6818 | 6885 | 13,707 |
| Macromolecule | 10,123 | 6864 | 3418 | 10,344 | 6841 | 6818 | 6845 | 13,620 |

*Table 1 continued on next page*

*Table 1 continued*

| | 2F2-210-1 | 2F2-210-2 | 2F2-210-3 | 2F2-210-4 | 2F2-210-5 | 2E10-247-2 | 2E10-247-3 | 2E10-247-4 |
|---|---|---|---|---|---|---|---|---|
| Water | 433 | 8 | 177 | 160 | 100 | - | 34 | 87 |
| Heteroatom | - | - | - | - | - | - | 6 | - |
| R S$_{work}$/R¶$_{free}$ | 17.9/22.0 | 20.2/24.9 | 18.6/23.5 | 18.6/22.8 | 19.2/23.8 | 18.0/21.0 | 20.7/23.9 | 20.9/24.6 |
| *Rms deviations from ideality* | | | | | | | | |
| Bond lengths (Å) | 0.007 | 0.002 | 0.015 | 0.002 | 0.003 | 0.010 | 0.010 | 0.010 |
| Bond angle (°) | 0.87 | 0.51 | 1.33 | 0.57 | 0.71 | 1.39 | 1.25 | 1.22 |
| *Ramachandran plot* | | | | | | | | |
| Favored regions (%) | 97.5 | 97.9 | 97.3 | 97.9 | 98.6 | 94.6 | 97.1 | 97.5 |
| Allowed regions (%) | 2.3 | 2.1 | 2.7 | 1.9 | 1.4 | 5.4 | 2.9 | 2.5 |
| *B-factors (Å²)* | | | | | | | | |
| Wilson B-value | 39.1 | 64.7 | 35.5 | 47.0 | 50.0 | 91.0 | 42.1 | 54.7 |
| Average B-factors | 45.7 | 75.0 | 45.7 | 51.0 | 53.7 | 111.0 | 71.0 | 77.0 |
| Average macromolecule | 45.7 | 75.9 | 45.9 | 51.1 | 53.7 | 111.0 | 71.4 | 77.2 |
| Average heteroatom | - | - | - | - | - | - | 69.8 | - |
| Average water molecule | 44.0 | 61.4 | 41.1 | 41.9 | 54.3 | - | 37.8 | 43.3 |

*Values in parentheses refer to the highest resolution bin.

†R$_{merge}$ = $\sum hkl \sum i |Ihkl, i - <Ihkl>| / \sum hkl< Ihkl >$.

‡R$_{pim}$ = $\sum hkl [1/(N - 1)]1/2 \sum i |Ihkl, i - <Ihkl>| / \sum hkl< Ihkl >$.

§R$_{work}$ = $(\sum ||Fo| - |Fc||) / (\sum ||Fo|)$.

¶5% of data were used for the R$_{free}$ calculation.

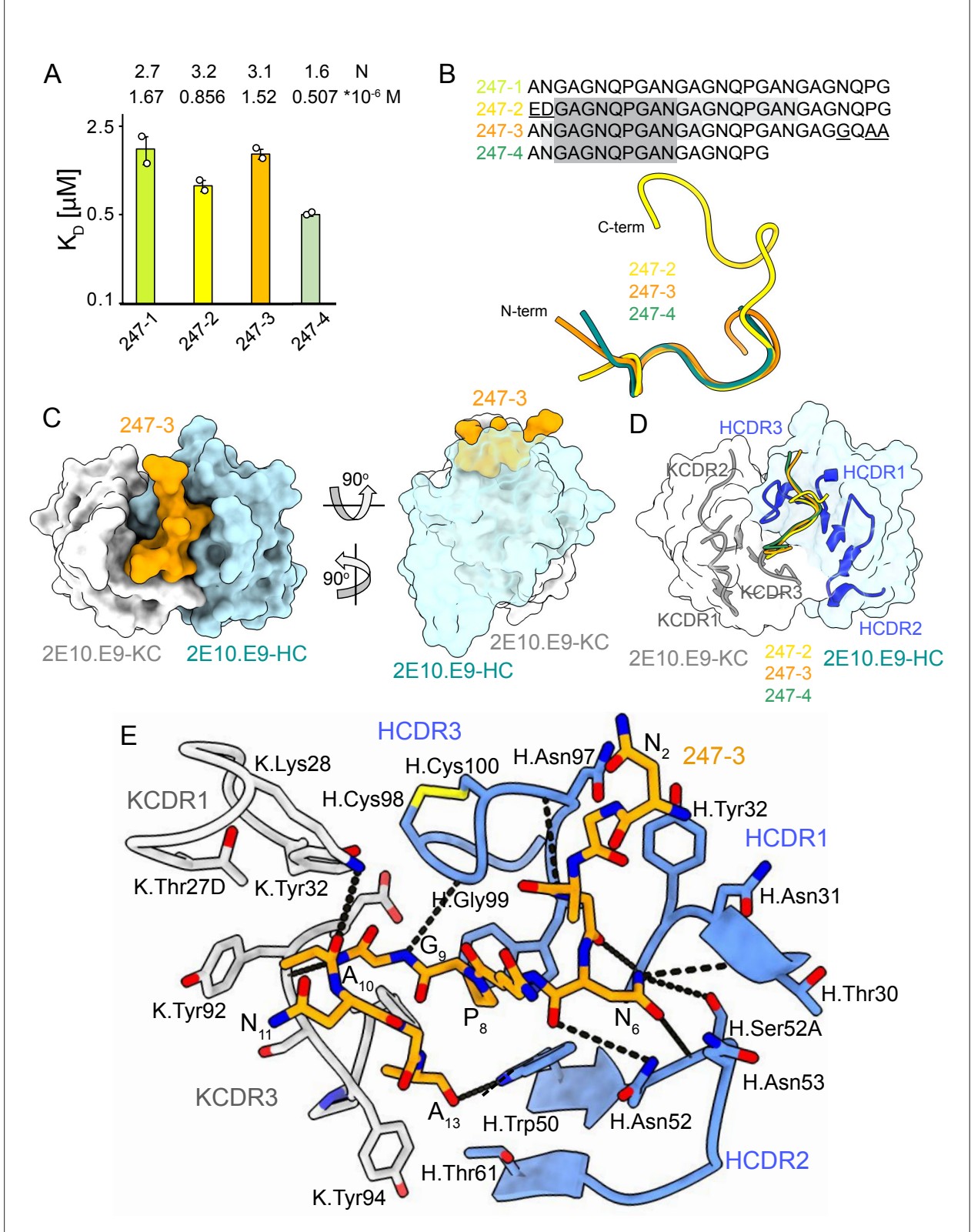

**Figure 4.** 2E10.E9 Fab binding to PvCSPvk247 repeat peptides. (**A**) Affinities of 2E10.E9 Fab for peptides 247-1, 247-2, 247-3, and 247-4 as measured by isothermal titration calorimetry (ITC). Open circles represent independent measurements. Mean binding constant ($K_D$) and binding stoichiometry (N) values are shown above the corresponding bar. Error bars represent SEM. (**B**) Upper panel: sequences of peptides used in ITC with variable residues underlined. Dark gray denotes the core epitope of the peptide resolved in all X-ray crystal structures, and light gray shading indicates residues resolved

*Figure 4 continued on next page*

*Figure 4 continued*

in the corresponding X-ray crystal structures. Bottom panel: comparison of the conformations of PvCSP247 peptides in X-ray crystal structures, with peptides 247-2, 247-3, and 247-4 depicted in yellow, orange, and teal, respectively. (**C**) Top and side views of the 247-3 peptide (orange) in the binding groove of the 2E10.E9 Fab shown as surface representation (heavy chain [HC] shown in blue and kappa chain [KC] shown in white). (**D**) Comparison of the conformations adopted by the core epitope of peptides 247-2, 247-3, and 247-4 when bound to 2E10.E9. (**E**) Detailed interactions between Fab 2E10.E9 and peptide 247-3. H-bonds are shown as black dashes, peptide 247-3 is shown in orange, HC is shown in green, and KC is shown in gray. The Fab residues are annotated with H or K letters to indicate heavy and kappa light chain, respectively.

The online version of this article includes the following figure supplement(s) for figure 4:

**Figure supplement 1.** Isothermal titration calorimetry (ITC) measurements of 2E10.E9 Fab binding to peptides 247-1 (**A**), 247-2 (**B**), 24-3 (**C**), and 247-4 (**D**).

**Figure supplement 2.** Stereo-image of the composite omit map electron density contoured at 1.2 sigma for peptides 247-2 (**A**), 247-3 (**B**), and 247-4 (**C**).

Fabs are mostly symmetric and involve mainly HCDR2 of both Fab A and B, as well as the KCDR3 of Fab A (*Figure 5A and B*). Fab-Fab interactions contribute 949 Å$^2$ of BSA (482 Å$^2$ on Fab A, 477 Å$^2$ on Fab B). The interaction between neighboring 2E10.E9 Fabs is stabilized by five H-bonds; three between HCDR2 residues of both Fab A and B and two additional H-bonds between K.Tyr94 of Fab A, and H.Asn53 and H.Thr73 of Fab B (*Figure 5B*). Comparison of the 2E10.E9 variable gene sequences to the inferred germline precursor (IGHV9-3 and IGKV8-19) reveals that only one of the residues involved in Fab-Fab contacts has been somatically hypermutated (H.Ser52A), making this homotypic Fab-Fab interaction primarily germline-encoded in the context of binding its repeating epitope (*Figure 5C*).

## Central repeat flexibility upon binding of inhibitory antibodies to full-length PvCSP

Next, we characterized the binding of mAbs 2F2 and 2E10.E9 to full-length recombinant PvCSPvk210 and PvCSPvk247. Both mAbs exhibit fast associations to their respective PvCSP sequences, as revealed in biolayer interferometry (BLI) experiments (*Figure 6A and C*). However, mAb 2E10.E9 displays a relatively fast dissociation compared to mAb 2F2, which contributes to the lower overall binding affinity of this mAb to PvCSPvk247 compared to a higher binding affinity for the mAb 2F2-PvCSPvk210 interaction. Next, ITC measurements indicated that, as expected, both 2F2 and 2E10.E9 Fabs bind PvCSP with high stoichiometry indicative of multiple Fab copies interacting with a single PvCSP molecule. 2F2 Fab recognizes PvCSPvk210 with ~10 times higher affinity (0.242 µM) compared to the 2E10.E9 Fab-PvCSPvk247 interaction (2.21 µM), corroborating the binding kinetics data (*Figure 6B and D*). Size-exclusion chromatography coupled with multiangle light scattering (SEC-MALS) characterization of the Fab-PvCSP complexes revealed high binding stoichiometry with a molecular weight of ~522 kDa and ~463 kDa for the 2F2 Fab-PvCSPvk210 and 2E10.E9 Fab-PvCSPvk247 complexes, respectively (*Figure 6E*). These sizes correspond to approximately 10 2F2 Fab's bound to one molecule of PvCSPvk210 and to approximately 9 2E10.E9 Fab's bound to one molecule of PvCSPvk247. Although both ITC and SEC-MALS confirm the assembly of large complexes formed by multiple Fab's binding to one PvCSP molecule, the exact Fab:PvCSP stoichiometry that ensues from these independent analyses is slightly different between the two techniques, which we attribute to the difficulty in obtaining precise concentration measurements for recombinant PvCSP and to the two experiments being performed at distinct concentrations.

To investigate a possible structural ordering of the PvCSP central repeat as might be induced by the binding of multiple 2F2 and 2E10.E9 Fab's, we performed negative stain electron microscopy (NS EM) and electron cryomicroscopy (cryo-EM) analyses of the SEC-purified Fab-PvCSP complexes (*Figure 6—figure supplement 1*). 2D class average images from the negative stain micrographs of the 2E10.E9 Fab-PvCSPvk247 complex revealed multiple 2E10.E9 Fabs spaced tightly against each other (*Figure 6—figure supplement 1A*, left panels). However, cryo-EM analysis of the same 2E10.E9 Fab-PvCSPvk247 complex indicated that Fab 2E10.E9 does not form regular, spiral assemblies with CSP, presumably because this type of complex would not accommodate the symmetric, head-to-head interactions between 2E10.E9 Fabs that were observed in the 2E10.E9 Fab-247-2 peptide crystal structure (*Figure 5*, *Figure 6—figure supplement 1A*, right panels).

Interestingly, 2D classes from the negative stain micrographs of the 2F2 Fab-PvCSPvk210 complex indicated multiple conformational states, and thereby suggest that the PvCSPvk210 central repeat

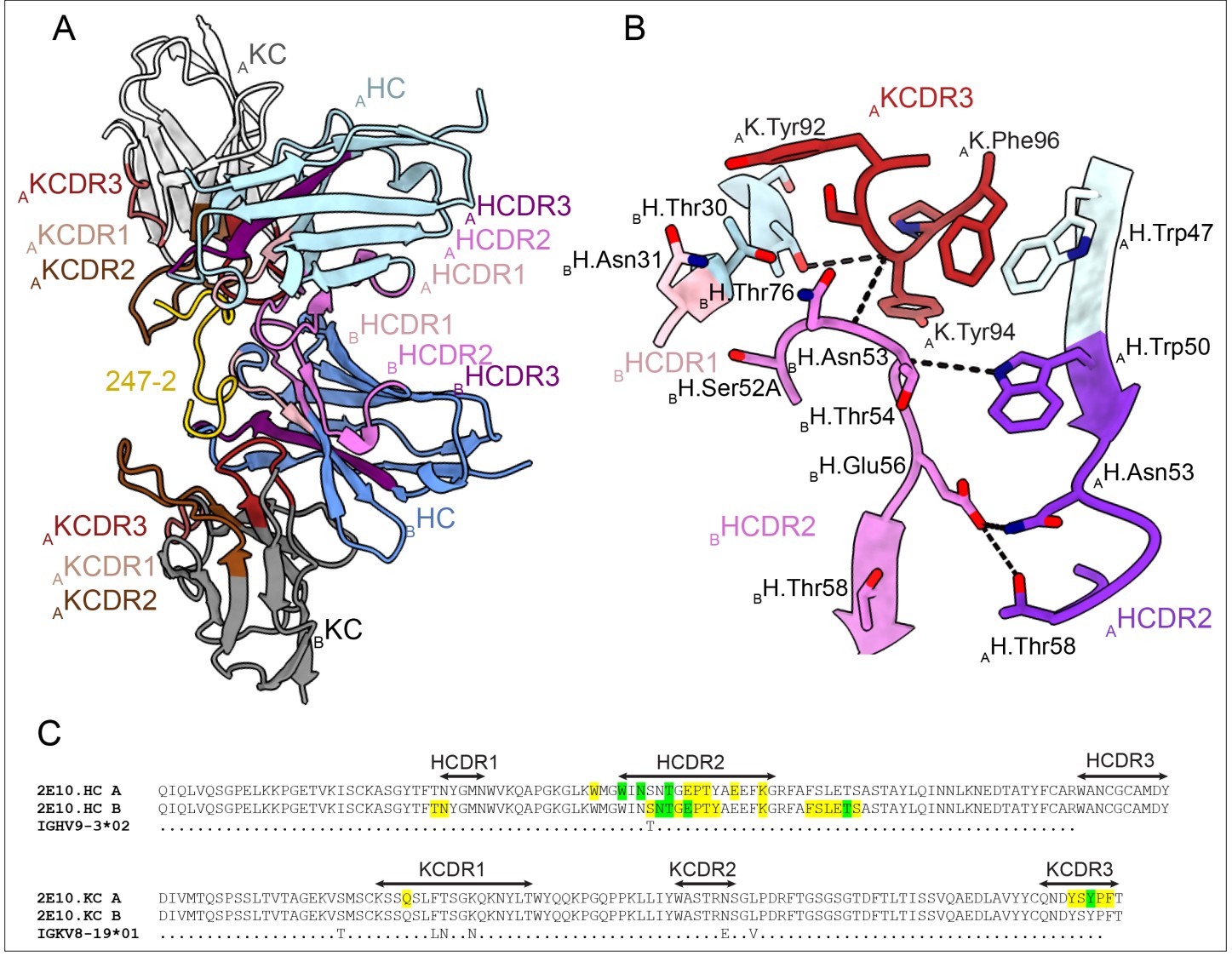

**Figure 5.** Homotypic Fab-Fab interactions in Fab 2E10.E9-247-2 peptide complex. (**A, B**) 2E10.E9 Fabs that simultaneously recognize the 247-2 peptide contact each other through an interface consisting of mainly of heavy chain complementarity-determining region (HCDR)2 of both Fab A and B, as well as kappa chain complementarity-determining region (KCDR)3 of Fab A. The heavy chain (HC) of Fab A and B is colored light and dark blue, respectively. The kappa chain (KC) of Fab A and B is colored light and dark gray, respectively. HCDR1, 2, and 3 are colored light pink, dark pink, and purple, respectively. KCDR1, 2, and 3 are shown in light brown, dark brown, and red, respectively. The 247-2 peptide is depicted in yellow. Black dashed lines denote H-bonds. Residues forming Fab-Fab contacts are labeled with the position of the Fab (A or B) indicated in subscript. (**C**) Sequence alignment of monoclonal antibody (mAb) 2E10.E9 with its inferred germline precursor. Yellow highlight: residues involved in homotypic interactions; green highlight: residues involved in homotypic interactions that form H-bonds.

region remains largely flexible upon binding by 2F2 Fabs (*Figure 6—figure supplement 1B*, left panels). Cryo-EM analysis of the same 2F2 Fab-PvCSPvk247 complex showed similar conformational heterogeneity (*Figure 6—figure supplement 1B*, right panels). These results suggest that the 2F2 Fabs may not be stabilized by appreciable inter-Fab contacts as observed for 2E10.E9 Fabs. To gain a better understating of the conformational flexibility observed for the 2F2 Fab-PvCSPvk210 complex, we collected NS EM data of the 2F2 Fab bound to a PvCSPvk210-derived peptide of sufficient length to accommodate binding of two Fabs (peptide 210-10, *Supplementary file 1*). EM class averages from the micrographs of the 2F2 Fab-210-10 peptide complex displayed high variability in the angles between the two Fabs, ranging from ~30° to ~170° (*Figure 7A*). These data indicate that the complex is highly flexible, and that the two Fabs are likely not forming extensive stabilizing inter-Fab homotypic contacts upon binding the repeat peptide. This mode of binding is in contrast to other Fab-peptide

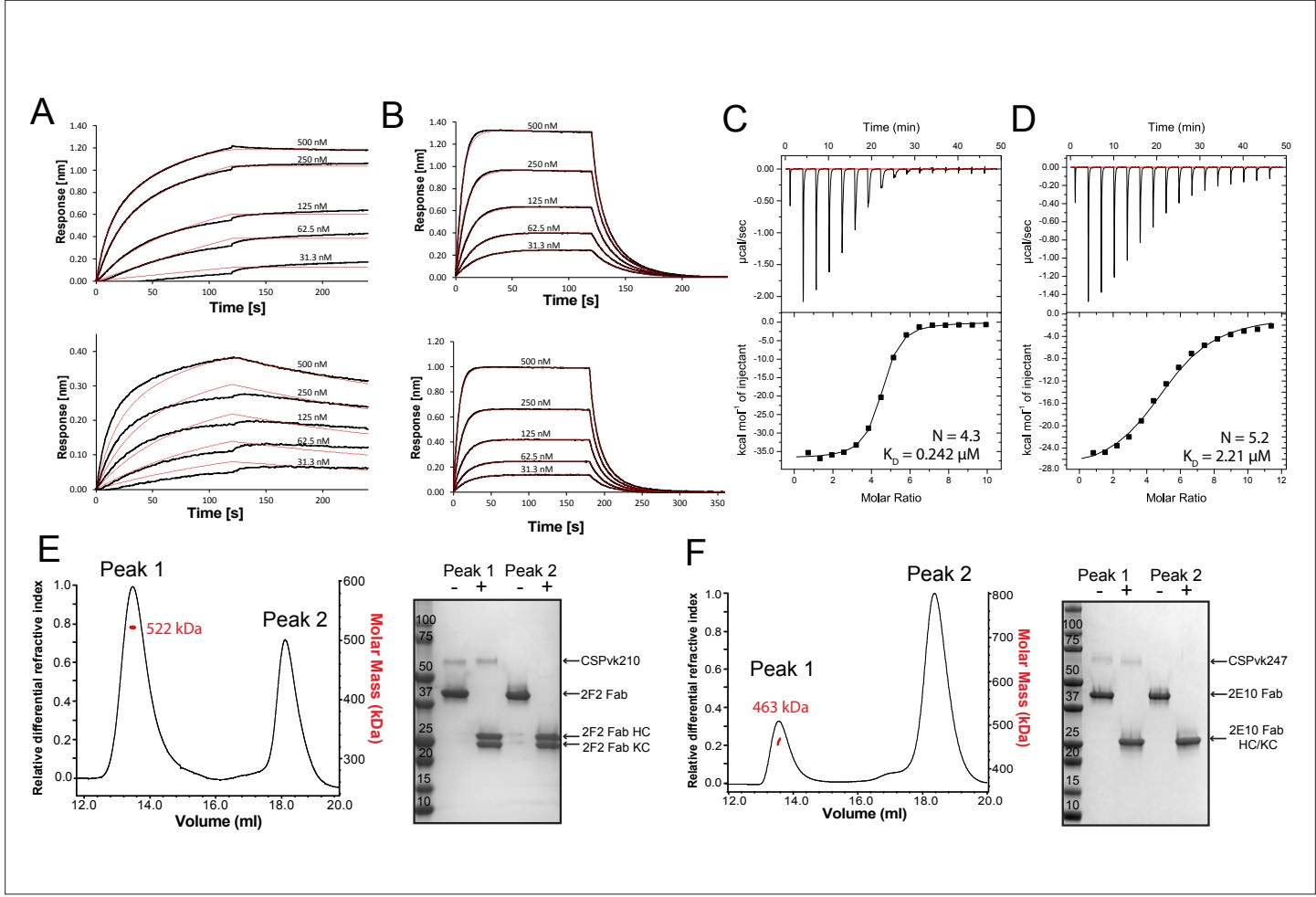

**Figure 6.** Binding of 2F2 and 2E10.E9 to full-length PvCSPvk210 and PvCSPvk247. Binding kinetics of twofold dilutions of 2F2 IgG and Fab (**A**, upper panel and lower panel, respectively) to PvCSPvk210, and 2E10.E9 IgG and Fab (**B**, upper panel and lower panel, respectively) to PvCSPvk247, as measured by biolayer interferometry (BLI). Representative sensograms are shown in black and 2:1 model best fits in red. Data shown are representative of three independent measurements. Isothermal titration calorimetry (ITC) analysis of 2F2 Fab binding to PvCSPvk210 (**C**) and 2E10.E9 Fab binding to PvCSPvk247 (**D**) at 25°C. (**C**, top panels): raw data of PvCSPvk210 (5 µM) in the sample cell titrated with 2F2 Fab (240 µM) in the syringe. (**D**, top panels): raw data of PvCSPvk247 (5 µM) in the sample cell titrated with 2E10.E9 Fab (400 µM) in the syringe. (**C, D**, bottom panel): plot and trendline of heat of injectant corresponding to the raw data. Results from size-exclusion chromatography coupled with multiangle light scattering (SEC-MALS) for the Fab 2F2-PvCSPvk210 sample (**E**, left panel) and 2E10.E9 Fab-PvCSPvk247 (**F**, left panel) sample, where the Fabs are in molar excess. Measurement of the molar mass of the eluting complex is shown as a red line. Mean molar mass is indicated. SDS-PAGE analysis of resulting peaks 1 and 2 for 2F2 Fab-PvCSPvk210 (**E**, right panel) and 2E10.E9 Fab-PvCSPvk247 (**F**, right panel) samples from SEC-MALS. Each peak was sampled in reducing and nonreducing conditions as indicated by + and –, respectively.

The online version of this article includes the following source data and figure supplement(s) for figure 6:

**Source data 1.** SDS-PAGE analysis of 2F2 Fab-PvCSPvk210 and 2E10.E9 Fab-PvCSPvk247 complexes.

**Figure supplement 1.** Negative stain electron microscopy (NS EM) and electron cryomicroscopy (cryo-EM) analysis of 2E10.E9 Fab-PvCSPvk247 (**A**) and 2F2 Fab-PvCSPvk210 (**B**) complexes.

complexes known to form extensive homotypic interactions (2E10.E9 Fab-247-2 [*Figure 7B*], 3D11 Fab-NPNDx2 [*Kucharska et al., 2020*; *Figure 7C*], and 1210 Fab-NANP$_5$ [*Imkeller et al., 2018*; *Figure 7D*, *Supplementary file 1*]). Complexes that form homotypic contacts showed lower flexibility than the 2F2 Fab-210-10 peptide complex, with either one or two distinct 3D classes present.

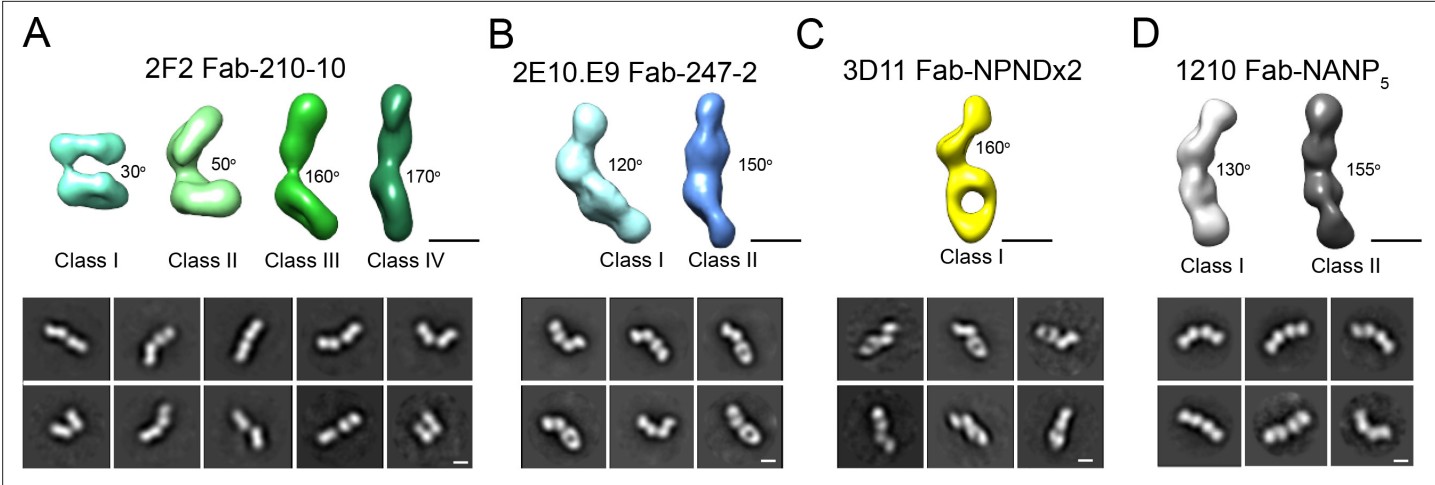

**Figure 7.** Evaluation of flexibility for various Fab-CSP peptide complexes by negative stain electron microscopy. Refined 3D classes (upper panels) and representative 2D class averages (bottom panels) of (**A**) 2F2 Fab-210-10 peptide, (**B**) 2E10.E9 Fab-247-2 peptide, (**C**) 3D11 Fab-NPNDx2 peptide (PPPPNPND)$_3$ (**Kucharska et al., 2020**), and (**D**) 1210 Fab-(NANP)$_5$ peptide (**Imkeller et al., 2018**) complexes. The approximate angle between adjacent Fabs in each class is indicated. Scale bars on 3D and 2D classes: 50 nm.

## Discussion

Previous knowledge of inhibitory mAbs against the PfCSP repeats suggest that an in-depth molecular understanding of PvCSP could facilitate the design of next-generation Pv biomedical interventions. Indeed, molecular characterization of hundreds of human mAbs induced either by natural infection (**Triller et al., 2017**), Pf sporozoite immunization (**Imkeller et al., 2018**; **Kisalu et al., 2018**; **Murugan et al., 2020**; **Tan et al., 2018**; **Wang et al., 2020**), or RTS,S/AS01 vaccination (**Oyen et al., 2017**; **Pholcharee et al., 2021**) revealed important insights into vaccine design and antibodies as prophylactics. These include preferential mAb binding to the conserved core epitope (N/D)PNANPN(V/A) (**Murugan et al., 2020**) and potential correlations of protection with binding affinity and recognition of epitopes with secondary structural motifs of type I β- and pseudo 3$_{10}$-turns (**Pholcharee et al., 2021**). Moreover, a subset of potent mAbs elicited by whole sporozoite vaccination was shown to not only bind to NANP repeats, but also to a junctional epitope positioned between the N terminus and the central repeat domain of PfCSP, which is absent in the RTS,S vaccine (**Kisalu et al., 2018**; **Tan et al., 2018**; **Wang et al., 2020**). Our understanding of CSP targeting by neutralizing mAbs is however almost exclusively limited to PfCSP. Despite the prominence of Pv malaria morbidity worldwide, a molecular understanding of PvCSP and how its central repeat is recognized by inhibitory antibodies has remained scarce.

Here, we performed CD spectroscopy and MD simulations on PvCSP repeat peptides to better understand the unliganded structure of this inhibitory antibody target. Our data show that all analyzed peptides are disordered in solution and any secondary structure observed is local and transient. This result is similar to what has been previously described for the Pf and Pb CSP repeat region (**Dyson et al., 1990**; **Kucharska et al., 2020**). Interestingly, the sequence composition of the CSP repeats from these three *Plasmodium* species is substantially different; whereas PfCSP and PbCSP are asparagine and proline-rich, PvCSP is predominantly alanine and glycine-rich. In addition, the repeating modulus for PfCSP and PbCSP contains four amino acids, whereas the repeating modulus of PvCSP contains nine amino acids. These comparisons underscore a diversity of amino acid features associated with low structural propensity in repeating sequences.

Interestingly, calculations of elastic modulus demonstrated that PvCSP peptides behave like harmonic springs, meaning that the stretching or compressing of the peptides is proportional to the force applied to them. Elastic properties are well characterized in a vast array of proteins of different functions, including elastin, spider silk, or mussel byssus (**Gosline et al., 2002**). The high degree of conformational disorder and the resulting elastic properties of PvCSP peptides likely stem from the combination of the low complexity periodic nature of the sequence as well as the particular amino acid composition of this region. A high combined proline and glycine content was shown to distinguish

the sequence of self-assembled elastomeric proteins such as elastin, resilin, and elastic spider silks, amongst many others, from that of proteins prone to amyloid formation (*Rauscher et al., 2006*). Both proline and glycine are 'amyloid breakers' because of their low propensity to form regular secondary structure such as the extended β-sheets found at the core of amyloid fibrils (*Parrini et al., 2005*; *Williams et al., 2004*). As such, a high proline and glycine content enables self-assembled elastomers to avoid the amyloid fate while remaining disordered even in the assembled or phase-separated state. In turn, high structural disorder enables these proteins to function as entropic springs, whereby the large conformational entropy of the polypeptide chain provides at least part of the driving force for elastic recoil. PvCSP peptides have glycine and proline contents of ~22–33% and ~7–11%, respectively, and thus are predicted to fall within the transition region separating amyloidogenic from elastomeric proteins, the latter of which include elastin domains and insect resilin (*Rauscher et al., 2006*). Moreover, the elastic modulus of PvCSP peptides (~3–5 cal/(mol Å$^2$)) is commensurate with the elastic modulus of peptides modeled after human elastin (~9 cal/(mol Å$^2$)), which requires extensibility and elasticity for its physiological function (*Reichheld et al., 2021*). This analogy supports the observation that CSP is inherently elastic and suggests that achieving a high degree of conformational disorder may be an essential aspect of CSP function on the surface of sporozoites. Interestingly, recent in vitro experiments demonstrated that PbCSP repeats also have elastic properties, which are lost when the repeat sequence is scrambled (*Balaban et al., 2021*). The link between the elasticity of CSP repeats and CSP function is still not completely understood; yet, it appears that biophysical properties of the CSP repeats are necessary for maintaining sporozoite motility (*Balaban et al., 2021*; *Coppi et al., 2011*), even though CSP itself is not a motor protein.

The presence of disorder in all analyzed PvCSP repeat peptide structures was also appreciated in our X-ray crystallography data, where inhibitory mAbs 2F2 and 2E10.E9 were found to recognize their epitopes in an induced coil conformation, with only one turn of a $3_{10}$-helix and isolated turns as secondary structure motifs observed in both instances. Binding of the CSP repeat by inhibitory antibodies has been shown to induce a range of conformations in this intrinsically disordered region for Pf and Pb, with the occasional presence of type I β-turns and pseudo $3_{10}$-turns often but not always linked with antibody-mediated protection (*Imkeller et al., 2018*; *Kisalu et al., 2018*; *Kucharska et al., 2020*; *Murugan et al., 2020*; *Oyen et al., 2017*; *Pholcharee et al., 2020*; *Pholcharee et al., 2021*; *Scally and Julien, 2018*; *Tan et al., 2018*; *Triller et al., 2017*).

PfCSP- and PbCSP-reactive antibodies have been shown to cross-react to a varying extent with the predominant repeat motif and neighboring sequences of subtle variance within the repeat region, for example, the PfCSP junction (*Julien and Wardemann, 2019*; *Kisalu et al., 2018*; *Murugan et al., 2020*; *Scally et al., 2018*; *Tan et al., 2018*). Cross-reactivity is a feature of human antibodies encoded by various Ig-gene combinations and was also observed in murine antibodies against the PfCSP repeat (mAb 2A10; *Zavala et al., 1983*) and against the PbCSP repeat (3D11; *Kucharska et al., 2020*; *Yoshida et al., 1980*). Our data indicate that both mAbs 2F2 and 2E10.E9 are cross-reactive as they bind to different repeat motifs of their respective PvCSP variants with similar affinity and in identical conformations. mAbs 2F2 and 2E10.E9 are, however, not cross-reactive to different strains of Pv (*Nardin et al., 1982*) due to the significantly different CSP sequences of strains PvCSPvk210 and PvCSPvk247. Although both mAbs 2F2 and 2E10.E9 are inhibitory, only mAb 2F2 was reported to induce aberrations in size and in DNA content of sporozoites, indicating a distinct mechanism of attenuation (*Roth et al., 2018*), possibly due to the higher affinity to PvCSP and slower dissociation rate of mAb 2F2 compared to mAb 2E10.E9.

Both mAbs 2F2 and 2E10.E9 recognize their respective epitopes using germline-encoded and somatically hypermutated residues. The difference in affinities between mAbs 2E10.E9 and 2F2 and their respective antigen might partially stem from the low lumber of somatic hypermutations of mAb 2E10.E9 (1 in HC, 6 in KC) compared to 2F2 (14 in HC, 4 in KC). Specifically, mAbs 2F2 and 2E10.E9 use nine and six germline-encoded aromatic residues, respectively, to mediate contacts with their core epitopes. Germline-encoded mAb 2E10.E9 residue H.Trp50 forms extensive van der Waals interactions with $N_6$ and $P_8$ residues, analogous to *IGHV3-33* germline-encoded H.Trp52 in several human mAbs, including mAbs MGG4 (*Tan et al., 2018*), 1210 (*Imkeller et al., 2018*), 311 (*Oyen et al., 2017*), and other antibodies induced by sporozoite immunization (*Murugan et al., 2020*) or RTS,S vaccination (*Pholcharee et al., 2021*; *Figure 3—figure supplement 3D and F*). As both human and murine mAbs appear to depend on germline-encoded aromatic residues for CSP recognition, it is likely that

CSP repeats prime the mammalian immune system to select antibodies from germline genes with already optimally positioned aromatic residues.

MAb 2E10.E9 displays homotypic Fab-Fab interactions, as was previously described for mAbs 1210 (*Imkeller et al., 2018*), 311 (*Oyen et al., 2018*), and 239 and 399 (*Pholcharee et al., 2021*) against PfCSP and murine mAb 3D11 against PbCSP (*Kucharska et al., 2020*). 2E10.E9 Fabs interact in a head-to-head binding mode, forming symmetric interactions similar to those observed in the case of antibodies 1210 (*Imkeller et al., 2018*) and 399 (*Pholcharee et al., 2021*). Interestingly, mAb 2E10. E9 does not appear to bind longer PvCSPvk247 peptides with higher affinity compared to shorter PvCSPvk247 peptides (*Figure 3A*), in contrast to what has been described for other mAbs that recognize the PfCSP and PbCSP repeats and form homotypic interactions (*Imkeller et al., 2018*; *Kucharska et al., 2020*; *Pholcharee et al., 2021*). Unlike mAbs 1210 (*Imkeller et al., 2018*) and 3D11 (*Kucharska et al., 2020*), the residues forming Fab-Fab contacts in mAb 2E10.E9 are almost exclusively germline-encoded, similar to mAb 399 (*Pholcharee et al., 2021*). Combined, these results suggest that forming Fab-Fab interactions is ubiquitous among anti-CSP mAbs targeting different *Plasmodium* species. Analysis of human mAbs isolated after immunization with whole Pf sporozoites indicated that NANP homotypic antibody interactions promote activation and strong clonal expansion of PfCSP-reactive B-cells (*Imkeller et al., 2018*). It appears that homotypic interactions may emerge from B-cell receptor clustering on the surface of B cells (*Imkeller et al., 2018*); however, a direct link between homotypic interactions in the context of soluble antibodies and sporozoite inhibition is still to be established. mAbs 2F2 and 2E10.E9 recognize core PvCSP epitopes of 10 and 8 residues, respectively. Interestingly, potent inhibitory mAbs against PfCSP and PbCSP typically recognize epitopes of similar length (8–10 residues; *Imkeller et al., 2018*; *Kucharska et al., 2020*; *Murugan et al., 2020*; *Pholcharee et al., 2021*), despite shorter repeating motifs for PfCSP and PbCSP (4-aa) compared to PvCSP (9-aa). Given the similar antibody epitope lengths, in addition to similar structural propensities for underlying antigenic motifs and induction of antibody homotypic interactions when recognizing closely spaced repeating epitopes, we suggest that similar CSP-based vaccine design approaches could be applied across different *Plasmodium* species. Nevertheless, the lack of *Plasmodium* species cross-reactivity for the potent inhibitory mAbs described thus far suggests that different pre-erythrocytic vaccines or a multicomponent CSP-based vaccine will likely be required for broad coverage against the different *Plasmodium* species causative of human malaria morbidity and mortality worldwide.

# Materials and methods

**Key resources table**

| Reagent type (species) or resource | Designation | Source or reference | Identifiers | Additional information |
|---|---|---|---|---|
| Recombinant DNA reagent | pcDNA3.4-2F2 Fab HC (plasmid) | This paper | N/A | 2F2 Fab heavy chain gene in pcDNA3.4 TOPO vector |
| Recombinant DNA reagent | pcDNA3.4-2F2 KC (plasmid) | This paper | N/A | 2F2 light chain gene in pcDNA3.4 TOPO vector |
| Recombinant DNA reagent | pcDNA3.4-2E10.E9 Fab HC (plasmid) | This paper | N/A | 2E10.E9 Fab heavy chain gene in pcDNA3.4 TOPO vector |
| Recombinant DNA reagent | pcDNA3.4-2E10.E9 KC (plasmid) | This paper | N/A | 2E10.E9 light chain gene in pcDNA3.4 TOPO vector |
| Recombinant DNA reagent | pcDNA3.4-PvCSPvk210-His6x (plasmid) | This paper | N/A | PvCSPvk210 gene with His tag in pcDNA3.4 TOPO vector |
| Recombinant DNA reagent | pcDNA3.4-PvCSPvk247-His6x (plasmid) | This paper | N/A | PvCSPvk247 gene with His tag in pcDNA3.4 TOPO vector |
| Cell line (*Homo sapiens*) | FreeStyle 293F cells | Thermo Fisher Scientific | Cat# R79007 | |

*Continued on next page*

*Continued*

| Reagent type (species) or resource | Designation | Source or reference | Identifiers | Additional information |
|---|---|---|---|---|
| Cell line (*Mus musculus*) | 2F2 hybridoma cell line | *Nardin et al., 1982*, Alan Cochrane, unpublished results | BEI Resources #MRA-184; RRID:CVCL_A7VR | |
| Cell line (*M. musculus*) | 2E10.E9 hybridoma cell line | *Nardin et al., 1982*, Alan Cochrane, unpublished results | BEI Resources #MRA-185; RRID:CVCL_A7VT | |
| Chemical compound, drug | Gibco FreeStyle 293 Expression Medium | Thermo Fisher Scientific | Cat# 12338026 | |
| Chemical compound, drug | Gibco Hybridoma-SFM | Thermo Fisher Scientific | Cat# 12045076 | |
| Chemical compound, drug | FectoPRO DNA Transfection Reagent | VWR | Cat# 10118-444 | |
| Chemical compound, drug | Fetal bovine serum | Thermo Fisher Scientific | Cat# 12483-020 | |
| Antibody | 2F2 IgG (mouse monoclonal) | *Nardin et al., 1982*, Alan Cochrane, unpublished results | N/A | Purified from 2F2 hybridoma cell line; see Materials and methods |
| Antibody | 2E10.E9 IgG (mouse monoclonal) | *Nardin et al., 1982*, Alan Cochrane, unpublished results | N/A | Purified from 2E10.E9 hybridoma cell line; see Materials and methods |
| Biological sample (*Carica papaya*) | Papain from papaya latex | Sigma-Aldrich | Cat# P4762 | |
| Peptide, recombinant protein | 1210 Fab | *Imkeller et al., 2018* | N/A | See Materials and methods for concentrations and masses used, and buffer conditions |
| Peptide, recombinant protein | 3D11 Fab | *Kucharska et al., 2020* | N/A | See Materials and methods for concentrations and masses used, and buffer conditions |
| Peptide, recombinant protein | 2F2 Fab | This paper | N/A | See Materials and methods for concentrations and masses used, and buffer conditions |
| Peptide, recombinant protein | 2E10.E9 Fab | This paper | N/A | See Materials and methods for concentrations and masses used, and buffer conditions |
| Peptide, recombinant protein | 210-1 (GDRADGQ PAGDRADGQPA) | This paper | N/A | Derived from PvCSPvk210 repeat region |
| Peptide, recombinant protein | 210-2 (GDRAAGQ PAGDRAAGQPA) | This paper | N/A | Derived from PvCSPvk210 repeat region |
| Peptide, recombinant protein | 210-3 (GDRADGQP AGDRAAGQPA) | This paper | N/A | Derived from PvCSPvk210 repeat region |
| Peptide, recombinant protein | 210-4 (GDRAAGQ PAGDRADGQP) | This paper | N/A | Derived from PvCSPvk210 repeat region |
| Peptide, recombinant protein | 210-5 (GDRAAGQ PAGNGAGGQAA) | This paper | N/A | Derived from PvCSPvk210 repeat region |

*Continued on next page*

*Continued*

| Reagent type (species) or resource | Designation | Source or reference | Identifiers | Additional information |
|---|---|---|---|---|
| Peptide, recombinant protein | 210-6 (GDRADGQ PAGDRADGQ PAGDRADGQPA) | This paper | N/A | Derived from PvCSPvk210 repeat region |
| Peptide, recombinant protein | 210-7 (GDRAAGQ PAGDRAAGQ PAGDRAAGQPA) | This paper | N/A | Derived from PvCSPvk210 repeat region |
| Peptide, recombinant protein | 210-8 (GDRADGQ PAGDRAAGQ PAGDRADGQPA) | This paper | N/A | Derived from PvCSPvk210 repeat region |
| Peptide, recombinant protein | 210-9 (GDRAAGQ PAGDRAAGQ PAGNGAGGQAA) | This paper | N/A | Derived from PvCSPvk210 repeat region |
| Peptide, recombinant protein | 210-10 (GDRADGQ PAGDRADGQ PAGDRADGQ PAGDRADGQPA) | This paper | N/A | Derived from PvCSPvk210 repeat region |
| Peptide, recombinant protein | 247-1 (ANGAGNQ PGANGAGNQ PGANGAGNQPG) | This paper | N/A | Derived from PvCSPvk247 repeat region |
| Peptide, recombinant protein | 247-2 (EDGAGNQ PGANGAGNQ PGANGAGNQPG) | This paper | N/A | Derived from PvCSPvk247 repeat region |
| Peptide, recombinant protein | 247-3 (ANGAGNQ PGANGAGNQ PGANGAGGQAA) | This paper | N/A | Derived from PvCSPvk247 repeat region |
| Peptide, recombinant protein | 247-4 (ANGAGNQ PGANGAGNQPG) | This paper | N/A | Derived from PvCSPvk247 repeat region |
| Peptide, recombinant protein | NPNDx2 (PPPPNPNDP PPPNPNDP PPPNPND) | *Kucharska et al., 2020* | N/A | Derived from PbCSP ANKA repeat region |
| Peptide, recombinant protein | NANP$_5$ (NANPNAN PNANPNA NPNANP) | *Imkeller et al., 2018* | N/A | Derived from PfCSP NF54 repeat region |
| Software, algorithm | GROMACS 2016.5 | *Abraham et al., 2015*; *Berendsen et al., 1995* | https://manual. gromacs.org/ documentation/ 2016-current/ index.html; RRID:SCR_014565 | |
| Software, algorithm | CHARMM22* | *Best and Hummer, 2009*; *Best and Mittal, 2010*; *Lindorff-Larsen et al., 2012*; *MacKerell et al., 1998*; *Piana et al., 2011* | https://www.charmm. org/charmm/?CFID= 66837e22-4ee5- 47ba-bcbf-b4b385c2397 e&CFTOKEN=0; RRID:SCR_014892 | |

*Continued on next page*

*Continued*

| Reagent type (species) or resource | Designation | Source or reference | Identifiers | Additional information |
|---|---|---|---|---|
| Software, algorithm | LINCS | *Hess, 2008* | N/A | |
| Software, algorithm | Particle-Mesh Ewald algorithm | *Darden et al., 1993*; *Essmann et al., 1995* | N/A | |
| Software, algorithm | Parrinello–Rahman algorithm | *Parrinello and Rahman, 1981* | N/A | |
| Software, algorithm | VMD | *Humphrey et al., 1996* | https://www.ks.uiuc.edu/Research/vmd/; RRID:SCR_001820 | |
| Software, algorithm | Matplotlib | *Hunter, 2007* | https://matplotlib.org/; RRID:SCR_008624 | |
| Software, algorithm | MDTraj | *McGibbon et al., 2015* | https://www.mdtraj.org/1.9.5/index.html | |
| Software, algorithm | Octet Data Analysis Software 9.0.0.6 | ForteBio | https://www.fortebio.com/products/octet-systems-software | |
| Software, algorithm | MicroCal ITC Origin 7.0 Analysis Software | Malvern | https://www.malvernpanalytical.com/ | |
| Software, algorithm | ASTRA | Wyatt | https://www.wyatt.com/products/software/astra.html; RRID:SCR_016255 | |
| Software, algorithm | GraphPad Prism 8 | GraphPad Software | https://www.graphpad.com/; RRID:SCR_002798 | |
| Software, algorithm | SBGrid | SBGrid Consortium | https://sbgrid.org/; RRID:SCR_003511 | |
| Software, algorithm | cryoSPARC v2 | *Punjani et al., 2017* | https://cryosparc.com/; RRID:SCR_016501 | |
| Software, algorithm | Relion | *Scheres, 2012* | https://www3.mrc-lmb.cam.ac.uk/relion/; RRID:SCR_016274 | |
| Software, algorithm | XDS | *Kabsch, 2010* | http://xds.mpimf-heidelberg.mpg.de/; RRID:SCR_015652 | |
| Software, algorithm | Phaser | *McCoy et al., 2007* | https://www.phenix-online.org/; RRID:SCR_014224 | |
| Software, algorithm | Phenix (phenix.refine; phenix.real_space_refine) | *Adams et al., 2010* | https://www.phenix-online.org/; RRID:SCR_014224 | |
| Software, algorithm | UCSF Chimera | *Pettersen et al., 2004* | https://www.cgl.ucsf.edu/chimera/; RRID:SCR_004097 | |
| Software, algorithm | UCSF ChimeraX | *Goddard et al., 2018* | https://www.cgl.ucsf.edu/chimerax/; RRID:SCR_015872 | |
| Software, algorithm | Coot | *Emsley et al., 2010* | https://www2.mrc-lmb.cam.ac.uk/personal/pemsley/coot/; RRID:SCR_014222 | |

*Continued on next page*

*Continued*

| Reagent type (species) or resource | Designation | Source or reference | Identifiers | Additional information |
|---|---|---|---|---|
| Software, algorithm | PyMOL | The PyMOL Molecular Graphics System, version 1.8 Schrödinger, LLC. | https://pymol.org/2/#products; RRID:SCR_000305 | |
| Software, algorithm | PDBePISA | *Krissinel and Henrick, 2007* | https://www.ebi.ac.uk/pdbe/pisa/; RRID:SCR_015749 | |
| Software, algorithm | Stride | *Heinig and Frishman, 2004* | http://webclu.bio.wzw.tum.de/stride/ | |
| Other | Homemade holey gold grids | *Marr et al., 2014* | N/A | |
| Other | Homemade carbon grids | *Booth et al., 2011* | N/A | |

## CD spectroscopy

The lyophilized samples of peptides 210-1, 210-2, 210-3, 210-4, 210-5, 247-1, 247-2, and 247-3 (*Supplementary file 1*) were suspended in pH 7.4 PBS buffer and diluted to 0.1 mg/mL. Spectra were collected using a Jasco J-1500 Spectropolarimeter with a cuvette path length of 0.1 cm, 2 s response time, 1 nm bandwidth, and a 100 nm/min scanning speed. Three spectra were collected at room temperature for each peptide.

## MD simulations

GROMACS 2016.5 (*Abraham et al., 2015*; *Berendsen et al., 1995*) was used to perform all-atom molecular simulations of the following seven peptides: 210-6, 210-7, 210-8, 210-9, 247-1, 247-2, and 247-3 (*Supplementary file 1*). The CHARMM22* (*Best and Hummer, 2009*; *Best and Mittal, 2010*; *Lindorff-Larsen et al., 2012*; *MacKerell et al., 1998*; *Piana et al., 2011*) force field and the CHARMM-modified TIP3P (TIPS3P) explicit water model (*Jorgensen et al., 1983*) were used for all simulations. PyMOL (*Schrödinger, 2015*) was used to design the peptides with acetylated N-terminus and amidated C-terminus. The peptides were first collapsed from their arbitrary extended state without any solvent (in vacuo) under NVT conditions. The last conformations from simulations in vacuo were used to initiate equilibrium simulations in water, with 20 replicas for each peptide.

Peptides were solvated with water and 0.15 M NaCl in a rhombic dodecahedral box, with a side length of 4.0 nm. Periodic boundary conditions were applied, and energy minimization was carried out using the steepest descent algorithm. Lennard–Jones and short-range electrostatic interactions were computed with a cutoff of 9.5 Å. Long-range electrostatic interactions were computed with Particle-Mesh Ewald summation (*Darden et al., 1993*; *Essmann et al., 1995*), using a fourth-order interpolation and a grid spacing of 1.2 Å. All bonds were constrained using the LINCS algorithm (*Hess, 2008*). The system was brought to the specified temperature of 300 K and pressure of 1 atm under NPT conditions. 10 ns NPT simulations were performed with velocity-rescaling temperature coupling (*Bussi et al., 2007*) and Berendsen pressure coupling (*Berendsen et al., 1984*). Finally, simulations were carried out under NPT conditions with Parrinello–Rahman pressure coupling (*Parrinello and Rahman, 1981*) for 300 ns. The integration step was 2 fs, and atomic coordinates were recorded every 100 ps.

## MD simulation analyses

Visual molecular dynamics (VMD) (*Humphrey et al., 1996*) was used to create snapshots of peptides, while all plots were created with Matplotlib (*Hunter, 2007*). The first 225 ns were excluded for computation of secondary structure propensities per residue, as well as for H-bonding contact maps. Secondary structure was assigned using the Python package MDTraj (*McGibbon et al., 2015*), which uses the DSSP algorithm (*Nagy and Oostenbrink, 2014*). An in-house script was used to compute H-bonding contact maps. Forward peptide-peptide H-bonds form between C=O of residue $i$ and N–H of residue $i + n$. A forward H-bond was identified if the donor-acceptor distance ($r_{ON}$) and the

hydrogen-donor-acceptor angle ($\theta$) were less than 3.5 Å and 37° for n = 2 ($\gamma$-turn), 4.9 Å and 66° for n = 3 ($\beta$-turns), 4.5 Å and 60° for n = 4 ($\alpha$-turn) and 3.5 Å and 40° for n ≥ 5. A reverse turn is formed between N–H of residue $i$ and C=O of residue $i + n$, and identified if $r_{ON} < 3.5$ Å and $\theta < 60°$ (except for n = 3, for which $\theta < 40°$). From a histogram of equilibrium end-to-end distances ($d$ = distance between C$\alpha$ of the first and last residues), the probabilities at each given $d$ ($P(d)$) were used to compute a free energy profile ($\Delta G(d)$) of the peptide's end-to-end distances. The resulting potential of mean force (PMF) was then fitted to the quadratic function of elastic potential energy:

$$\Delta G(d) = -kBTlnP(d) \cong k(d - d_0)^2 \qquad (1)$$

where $k_B$ is the Boltzmann constant, $T$ is the absolute temperature, $k$ represents the stiffness or elastic modulus of the peptide, and $d_0$ is the equilibrium end-to-end distance.

## 2F2 and 2E10.E9 IgG expression and purification

Hybridoma cell lines for mAbs 2F2 and 2E10.E9 (BEI Resources MRA-184 and MRA-185, respectively) were cultured in Gibco Hybridoma-SFM (Thermo Fisher Scientific Cat#12045076) with 5–20% fetal bovine serum (Thermo Fisher Scientific Cat# 12483-020). After 5–7 days, cells were harvested and centrifuged. The supernatant containing 2F2 or 2E10.E9 IgG was purified via Protein G affinity chromatography (GE Healthcare) and size-exclusion chromatography (Superose 6 Increase 10/300 GL, GE Healthcare).

## 2E10.E9 IgG papain digestion

To obtain Fab, 2E10.E9 IgG was digested with papain (Sigma-Aldrich Cat#P4762) at a 5:1 molar ratio for 16 h at 37°C in PBS, 10 mM EDTA, 20 mM cysteine, pH 7.4. Fc, and non-digested IgG were removed via Protein A affinity chromatography (GE Healthcare). 2E10.E9 Fab was subsequently purified via size-exclusion chromatography (Superdex 200 Increase 10/300 GL, GE Healthcare), concentrated, and diluted to 10 mg/mL with peptide 247-4 to immediately use in crystallization experiments.

## 2F2 Fab and 2E10.E9 Fab expression and purification

Variable light and heavy chains of mAb 2F2 and 2E10.E9 antibody genes were sequenced from the hybridomas (BEI Resources MRA-184 and MRA-185, respectively; Applied Biological Materials Inc). Sequenced regions were gene synthesized and cloned (GeneArt) into custom pcDNA3.4 expression vectors immediately upstream of human Ig$\kappa$ and Ig$\gamma$1-C$_H$1 domains. pcDNA3.4-Fab KC and Fab HC plasmids were co-transfected into HEK 293F cells for transient expression using FectoPRO DNA transfection reagent (Polyplus). Cells were cultured in Gibco FreeStyle 293 Expression Medium for 6–7 days and subsequently purified via a combination of KappaSelect affinity chromatography (GE Healthcare), cation exchange chromatography (MonoS, GE Healthcare), and size-exclusion chromatography (Superdex 200 Increase 10/300 GL, GE Healthcare).

## Recombinant PvCSP expression and purification

Constructs of full-length PvCSPvk210 (GenBank accession number: AAA29526.1) and PvCSPvk247 (GenBank accession number: AAA29506.1) were gene synthesized and cloned (GeneArt) into pcDNA3.4 expression vectors with a His$_{6x}$ tag. The resulting pcDNA3.4-PvCSPvk210-His$_{6x}$ and pcDNA3.4-PvCSPvk247-His$_{6x}$ plasmids were transiently transfected in HEK 293F cells using FectoPRO DNA transfection reagent (Polyplus), cultured in Gibco FreeStyle 293 Expression Medium, and purified by HisTrap FF affinity chromatography (GE Healthcare) and size-exclusion chromatography (Superdex 200 Increase 10/300 GL, GE Healthcare).

## Cell lines

HEK 293F cells (Thermo Fisher Scientific 12338026) and mAb 2F2 and 2E10.E9 hybridoma cell lines (BEI Resources MRA-184 and -185, respectively) were authenticated and validated to be mycoplasma-free by their respective commercial entities.

## Biolayer interferometry

BLI (Octet RED96, ForteBio) experiments were conducted to determine the binding kinetics of 2F2 and 2E10.E9 IgG and Fab to recombinant PvCSPvk210 and PvCSPvk247. PvCSPvk210 or PvCSPvk247 was diluted to 10 μg/mL in kinetics buffer (PBS, pH 7.4, 0.01% [w/v] BSA, 0.002% [v/v] Tween-20) and immobilized onto Ni-NTA biosensors (ForteBio). Subsequently, biosensors were dipped into wells containing dilutions of either 2F2 or 2E10.E9 IgG or Fab in kinetics buffer. For measurement of the dissociation rate, tips were immersed back into kinetics buffer after association. All data were analyzed using ForteBio's Octet Data Analysis software 9.0.0.6, and curves were fitted to a 2:1 binding model given the presence of multiple epitopes of slightly different sequence composition for these mAbs within a single PvCSP molecule.

## Isothermal titration calorimetry

ITC experiments were performed with an Auto-iTC200 instrument (Malvern) at 25°C. Titrations were performed with 2F2 or 2E10.E9 Fab in the syringe in 15 successive injections of 2.5 μl. Full-length recombinant PvCSPvk210 and PvCSPvk247, and PvCSP-derived peptides (*Supplementary file 1*) were added to the calorimetric cell. All proteins and peptides were diluted in Tris-buffered saline (TBS; 20 mM Tris pH 8.0, and 150 mM NaCl). Full-length recombinant PvCSP was diluted to 5 μM and titrated with Fab at 240–400 μM. All PvCSP-derived peptides were diluted to 4–8 μM and titrated with 2F2 Fab at 100–125 μM or 2E10.E9 Fab at 180–220 μM. Experiments were performed at least in duplicates, and the mean and standard error of the mean are reported. ITC data were analyzed using the Micro-Cal ITC Origin 7.0 Analysis Software according to a 1:1 binding model.

## Size-exclusion chromatography-multiangle light scattering

Full-length recombinant PvCSPvk210 or PvCSPvk247 was complexed with a molar excess of 2F2 or 2E10.E9 Fab and loaded on a Superose 6 Increase 10/300 GL column (GE Healthcare) using an Agilent Technologies 1260 Infinity II HPLC coupled inline with the following calibrated detectors: (i) MiniDawn Treos MALS detector (Wyatt); (ii) quasi elastic light scattering (QELS) detector (Wyatt); and (iii) Optilab TreX refractive index (RI) detector (Wyatt). Data processing was performed using the ASTRA software (Wyatt).

## Crystallization and structure determination

Purified 2F2 Fab was concentrated and diluted to 5 mg/mL with peptides 210-1, 210-2, 210-3, 210-4, and 210-5 (*Supplementary file 1*) in a 1:3 molar ratio. The 2F2 Fab-peptide complexes were mixed in a 1:1 ratio with the following conditions to obtain crystals: 0.2 M ammonium sulfate, 25% (w/v) PEG 4000, 0.1 M sodium acetate, pH 4.6 (2F2 Fab-210-1); 0.1 M CHES, pH 9.6, and 20% (w/v) PEG 8000 (2F2 Fab-210-2); 0.2 M potassium acetate and 20% (w/v) PEG 3350 (2F2 Fab-210-3); 0.1 M sodium citrate, pH 5.0, and PEG 6000 (2F2 Fab-210-4); and 0.2 M disodium tartrate and 20% (w/v) PEG 3350 (2F2 Fab-210-5). Crystals appeared after 1–3 days and were cryoprotected in 15% (v/v) ethylene glycol (2F2 Fab-210-1 and 2F2 Fab-210-3), 20% (v/v) ethylene glycol (2F2 Fab-210-2 and 2F2 Fab-210-5) or 25% glycerol (2F2 Fab-210-4), before being flash-frozen in liquid nitrogen.

Purified 2E10.E9 Fab was concentrated and diluted to 10 mg/mL with peptide 247-4 in a 1:5 molar ratio and immediately used for setting up crystal trays. After complexing with peptides 247-2 or 247-3 (*Supplementary file 1*) in a 2:1 molar ratio, 2E10.E9 Fab-peptide complexes were purified by size-exclusion chromatography (Superdex 200 Increase 10/300 GL, GE Healthcare), concentrated, and diluted to 10 mg/mL. The 2E10.E9 Fab-peptide complexes were mixed in a 1:1 ratio with the following conditions to obtain crystals: 0.2 M potassium sodium tartrate and 20% (w/v) PEG 3350 (2E10.E9 Fab-247-4); 0.08 M sodium acetate, pH 4.6, 20% glycerol (v/v), and 1.6 M ammonium sulfate (2E10.E9 Fab-247-2); and 0.1 M sodium cacodylate, pH 6.5, 0.2 M sodium acetate, and 30% (w/v) PEG 8000 (2E10.E9 Fab-247-3). Crystals appeared after 1–5 days and were cryoprotected in 20% (v/v) glycerol (2E10.E9 Fab-247-4) or 20% ethylene glycol (2E10.E9 Fab-247-2) before being flash-frozen in liquid nitrogen. For 2E10.E9 Fab-247-3 crystals, no additional cryoprotectant was added.

Data were collected at the 23-ID-D or 23-ID-B beamlines at the Argonne National Laboratory Advanced Photon Source. All datasets were processed and scaled using XDS (*Kabsch, 2010*). The structures were determined by molecular replacement using Phaser (*McCoy et al., 2007*). Refinement of the structures was performed using phenix.refine (*Adams et al., 2010*) and iterations of

refinement using Coot (*Emsley et al., 2010*). Access to all software was supported through SBGrid (*Morin et al., 2013*). Prediction of secondary structure of the co-crystallized peptides was performed with Stride (*Heinig and Frishman, 2004*). Fab-peptide and Fab-Fab contacts were analyzed using the PDBePisa server (*Krissinel and Henrick, 2007*). The detection of intramolecular H-bonds in peptides was performed with PyMOL (*Schrödinger, 2015*).

## Negative stain EM

2F2 Fab-PvCSPvk210 or 2E10.E9 Fab-PvCSPvk247 complexes were purified on a Superose 6 Increase 10/300 GL column (GE Healthcare) and diluted to 50 µg/mL. 2F2 Fab-210-6, 2E10.E9 Fab-247-2, 3D11 Fab-NPNDx2, and 1210 Fab-NANP$_5$ peptide complexes were purified on a Superdex 200 Increase 10/300 GL column (GE Healthcare) and diluted to 25 µg/mL. Samples were deposited onto home-made carbon film coated grids and stained with 2% uranyl formate. Specimens were imaged with a FEI Tecnai T20 electron microscope operating at 200 kV with an Orius charge-coupled device (CCD) camera (Gatan Inc). A calibrated 34,483× magnification, resulting in a pixel size of 2.71 Å, was used for data collection. Particle selection, extraction, and three rounds of 2D classification with 50 classes were performed with Relion (*Scheres, 2012*) and cryoSPARC v2 (*Punjani et al., 2017*).

## Cryo-EM data collection and image processing

2F2 Fab-PvCSPvk210 and 2E10.E9 Fab-PvCSPvk247 complexes were purified via Superose 6 Increase 10/300 GL chromatography (GE Healthcare) and concentrated to 0.5 mg/mL. 3 µL of the sample was deposited on homemade holey gold grids (*Marr et al., 2014*), which were glow-discharged in air for 15 s before use. Samples were blotted for 12.0 s and subsequently plunge-frozen in a mixture of liquid ethane and propane (*Tivol et al., 2008*) using a modified FEI Vitrobot (maintained at 4°C and 100% humidity). Data collection was performed with a FEI Tecnai F20 microscope operated at 200 kV with a K2 camera (Gatan Inc). A calibrated 34,483× magnification, resulting in a pixel size of 1.45 Å, and defocus range between 1.5 and 2.8 µm were used for data collection. Exposures were fractionated as movies of 30 frames with a total exposure of 35 electrons/Å$^2$. A total of 269 movies were obtained for the 2E10.E9 Fab-PvCSPvk247 complex and 169 movies for the 2F2 Fab-PvCSPvk210 complex. Image processing was carried out in cryoSPARC v2 (*Punjani et al., 2017*). Initial specimen motion correction, exposure weighting, and CTF parameters estimation were done using patch-based algorithms. 100,133 and 87,287 particle images were extracted from micrographs of 2E10.E9 Fab-PvCSPvk247 and 2F2 Fab-PvCSPvk210 complex, respectively, and subjected to 4–5 rounds of 2D classification.

## Acknowledgements

We thank Dr. S Scally for his input during the course of this work. We are grateful to Dr. S Benlekbir for advice regarding specimen preparation. This work was supported by the CIFAR Azrieli Global Scholar program (JPJ), the Ontario Early Researcher Award program (JPJ), the Canada Research Chair program (JLR, JPJ), and the Canadian Institutes of Health Research (RP). IK and DI are supported by SickKids Restracomp Fellowships. This research was enabled in part by support provided by Compute Canada (https://www.computecanada.ca/). The ITC and BLI instruments were accessed at the Structural and Biophysical Core Facility, The Hospital for Sick Children, supported by the Canada Foundation for Innovation and Ontario Research Fund. X-ray diffraction experiments were performed at GM/CAAPS, which has been funded in whole or in part with federal funds from the National Cancer Institute (ACB-12002) and the National Institute of General Medical Sciences (AGM-12006). The Eiger 16M detector was funded by an NIH–Office of Research Infrastructure Programs High-End Instrumentation grant (1S10OD012289-01A1). This research used resources of the Advanced Photon Source, a U.S. Department of Energy (DOE) Office of Science user facility operated for the DOE Office of Science by Argonne National Laboratory under contract DE-AC02-06CH11357. The following reagents were obtained through BEI Resources, NIAID, NIH: MRA-184, Hybridoma 2F2 Anti-*Plasmodium vivax* Circumsporozoite Protein (CSP) and MRA-185, Hybridoma 2E10.E9 Anti-*Plasmodium vivax* Circumsporozoite Protein (CSP). X-ray crystallography structures are accessible from the Protein Data Bank under PDB IDs: 7RLV (2F2 Fab-210-1), 7RLW (2F2 Fab-210-2), 7RLX (2F2 Fab-210-3), 7RLY (2F2 Fab-210-4), 7RLZ (2F2 Fab-210-5), 7RM1 (2E10.E9 Fab-247-2), 7RM3 (2E10.E9 Fab-247-3), and 7RM0 (2E10.E9 Fab-247-4).

## Additional information

### Funding

| Funder | Grant reference number | Author |
| --- | --- | --- |
| Canada Research Chairs | | John L Rubinstein<br>Jean-Philippe Julien |
| Canadian Institutes of Health Research | | Régis Pomès |
| Sick Kids Foundation | Restracomp Fellowship | Iga Kucharska<br>Danton Ivanochko |
| Canadian Institute for Advanced Research | Azrieli Global Scholar program | Jean-Philippe Julien |
| Ontario Research Foundation | Early Researcher Award program | Jean-Philippe Julien |

The funders had no role in study design, data collection and interpretation, or the decision to submit the work for publication.

### Author contributions

Iga Kucharska, Conceptualization, Formal analysis, Funding acquisition, Investigation, Methodology, Project administration, Supervision, Validation, Visualization, Writing - original draft, Writing – review and editing; Lamia Hossain, Conceptualization, Formal analysis, Investigation, Methodology, Validation, Visualization, Writing - original draft, Writing – review and editing; Danton Ivanochko, Formal analysis, Funding acquisition, Methodology, Validation, Visualization, Writing – review and editing; Qiren Yang, Formal analysis, Investigation, Methodology, Writing – review and editing; John L Rubinstein, Funding acquisition, Methodology, Resources, Writing – review and editing; Régis Pomès, Conceptualization, Funding acquisition, Methodology, Project administration, Resources, Supervision, Writing – review and editing; Jean-Philippe Julien, Conceptualization, Funding acquisition, Investigation, Methodology, Project administration, Supervision, Writing – review and editing

### Author ORCIDs

Iga Kucharska http://orcid.org/0000-0001-6150-3419
John L Rubinstein http://orcid.org/0000-0003-0566-2209
Régis Pomès http://orcid.org/0000-0003-3068-9833
Jean-Philippe Julien http://orcid.org/0000-0001-7602-3995

### Decision letter and Author response

Decision letter https://doi.org/10.7554/eLife.72908.sa1
Author response https://doi.org/10.7554/eLife.72908.sa2

## Additional files

### Supplementary files

- Supplementary file 1. Summary of CSP-derived peptides used in this study.

- Supplementary file 2. Intramolecular H-bonds (3.0 Å cutoff) in PvCSP peptides observed in Fab-peptide co-crystal structures. No intramolecular H-bonds were detected for peptide 247-4.

- Transparent reporting form

### Data availability

X-ray crystallography structures are accessible from the Protein Data Bank under PDB IDs: 7RLV (2F2 Fab-210-1), 7RLW (2F2 Fab-210-2), 7RLX (2F2 Fab-210-3), 7RLY (2F2 Fab-210-4), 7RLZ (2F2 Fab-210-5), 7RM1 (2E10.E9 Fab-247-2), 7RM3 (2E10.E9 Fab-247-3), 7RM0 (2E10.E9 Fab-247-4).

The following datasets was generated:

| Author(s) | Year | Dataset title | Dataset URL | Database and Identifier |
|---|---|---|---|---|
| Kucharska I, Julien JP | 2022 | Antibody 2F2 in complex with P. vivax CSP peptide GDRADGQPAGDRADGQPA | https://www.rcsb.org/structure/7RLV | RCSB Protein Data Bank, 7RLV |
| Kucharska I, Julien JP | 2022 | Antibody 2F2 in complex with P. vivax CSP peptide GDRAAGQPAGDRAAGQPA | https://www.rcsb.org/structure/7RLW | RCSB Protein Data Bank, 7RLW |
| Kucharska I, Julien JP | 2022 | Antibody 2F2 in complex with P. vivax CSP peptide GDRADGQPAGDRAAGQPA | https://www.rcsb.org/structure/7RLX | RCSB Protein Data Bank, 7RLX |
| Kucharska I, Julien JP | 2022 | Antibody 2F2 in complex with P. vivax CSP peptide DRAAGQPAGDRADGQPA | https://www.rcsb.org/structure/7RLY | RCSB Protein Data Bank, 7RLY |
| Kucharska I, Julien JP | 2022 | Antibody 2F2 in complex with P. vivax CSP peptide GDRAAGQPAGNGAGGQAA | https://www.rcsb.org/structure/7RLZ | RCSB Protein Data Bank, 7RLZ |
| Kucharska I, Ivanochko D, Julien JP | 2022 | Antibody 2E10.E9 in complex with P. vivax CSP peptide ANGAGNQPGANGAGNQPG | https://www.rcsb.org/structure/7RM0 | RCSB Protein Data Bank, 7RM0 |
| Kucharska I, Ivanochko D, Julien JP | 2022 | Antibody 2F2 in complex with P. vivax CSP peptide EDGA GNQPGANGAGNQPGAN GAGNQPG | https://www.rcsb.org/structure/7RM1 | RCSB Protein Data Bank, 7RM1 |
| Kucharska I, Ivanochko D, Julien JP | 2022 | Antibody 2E10.E9 in complex with P. vivax CSP peptide ANGAGNQPGANGAGNQ PGANGAGGQAA | https://www.rcsb.org/structure/7RM3 | RCSB Protein Data Bank, 7RM3 |

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
