## [Editor Report]

This paper combines simulations and experimental biophysical approaches to probe the molecular basis of antibody recognition of key repeat peptides found on the surface of *Plasmodium vivax*, a parasite that causes malaria. Currently, we know surprisingly little about the mechanism by which antibodies recognize these peptides. Thus, this work provides crucial molecular insights into a recognition process of considerable biomedical significance, which might ultimately inform rational design of novel and more effective antimalarial vaccines.

---

## [Decision Letter]

**Decision letter after peer review:**

Thank you for submitting your article "Structural basis of *Plasmodium vivax* inhibition by antibodies binding to the circumsporozoite protein repeats" for consideration by *eLife*. Your article has been reviewed by 3 peer reviewers, one of whom is a member of our Board of Reviewing Editors, and the evaluation has been overseen by José Faraldo-Gómez as the Senior Editor. The following individual involved in review of your submission has agreed to reveal their identity: Niraj H Tolia (Reviewer #3).

Essential revisions:

Overall, the reviewers agree that this paper is strong. The science is rigorous and well described in the manuscript. One area that needs addressing is a more thorough discussion of the biological insights derived from these studies, which, as you know, is a critical component of articles published in *eLife*.

Additionally, the reviewers have made note of a number of specific issues that must be addressed in your revised submission (see below):*Reviewer #1:*

Plasmodium vivax (Pv), which causes Malaria, is decorated with surface receptors that contain central repeat proteins that antibodies can target to prevent Pv infection. Thus understanding the molecular basis of antibody recognition of the peptides in these repeats regions might help advance our basic science understanding and potentially aid in vaccine design. Here, Kucharska et al., used a battery of techniques to characterize the biophysical properties of two strains of Pv repeat peptides, VK210 and VK247, found in the central repeat regions of the Pv receptor protein, Circumsporozoite protein (CSP). A particular strength of this work is the diversity of biophysical techniques used to study the peptides in the absence and presence of inhibitory antibodies, which enabled the authors to provide a comprehensive picture of the biophysical properties of these peptides. However, lacking from the current account is an articulation of the biological insights derived from these studies.

1. In Figure 2a, the authors indexed the 210 peptides using letters (a-d), whereas, in other parts of the manuscript, they indexed the peptides using numbers (1-5). The authors should use consistent indexing.

2. Can the authors clarify the difference between peptides 210-1 and 210-2? In Figure 1, they appear to have the same sequence. Maybe a typo?

3. The authors report in Figure 1 CD results for the five 210 peptides. However, in Figure 2, they only show MD simulations for four of the 210 peptides. Can the authors comment on why?

4. The authors should comment on how the elastic moduli they compute compare with other peptides/proteins. For example, do the values fall within the range of what others have calculated for peptides with known elastic properties? This is important because when discussing the molecular basis of 2F2, the authors describe the peptides as elastic, as if self-evident. Are there any easy experiments that the authors can perform to test the MD prediction that these peptides are elastic?

5. The authors state that "2F2 Fab binds to all tested peptides containing two repeats with similar affinity, with a slight preference for peptide 210-1 (GDRADGQPAGDRADGQPA; 17.0 nM) 137 over 210-2 (GDRAAGQPAGDRAAGQPA; 37.5 nM) (Figure 3A). "

However, based on Figure 3a, it seems like the slight preference is force 210-3, not 210-1. Am I mistaken?

6. I know that Figure 3e is already a bit crowded, but the authors should consider annotating the image with the distances associated with the contacts they highlighted (dashed lines).

*Reviewer #2:*

The manuscript from Kucharska et al., investigates the structural basis for antibody recognition of the central repeat regions of the Plasmodium vivax (Pv) circumsporozoite protein (CSP). Similar studies by this group and others have been performed for the *Plasmodium falciparum* (Pf) CSP, but information on Pv has been lacking. Circular dichroism and MD studies indicate that the PvCSP central repeats are largely unstructured. ITC, BLI, X-ray crystallography and EM studies define the structural basis for the recognition of PvCSP repeats from strains VK210 and VK247 by two murine antibodies, 2F2 and 2E10.E9. In the crystal structure of 2E10.E9 in complex with a longer repeat peptide, homotypic 2E10.E9 Fab-Fab interactions are observed, as observed previously for potent antibodies against PfCSP. The EM studies demonstrate that binding of 2F2 or 2E10.E9 Fabs to PvCSP does not induce the central repeat region to form a spiral assembly, as observed for some of the potent PfCSP antibodies against.

The manuscript is clear, concise, and well written. Although I can not assess the molecular dynamics studies, the structural and biophysical experiments were rigorously performed, with excellent statistics for the 8 crystal structures. This manuscript should be of interest to those investigators involved in malaria vaccine development.*Reviewer #3:*

CSP is a prominent vaccine antigen that has been extensively characterized for *P. falciparum* malaria, but less is known about the *P. vivax* CSP and its potential for antibody-mediated protection. The authors propose that the CSP repeat regions are elastic and may provide a high degree of conformational flexibility relevant for CSP function. They further describe structural studies of the repeat region of *P. vivax* CSP in conjunction with rodent antibodies and show that antibodies that bind the CSP repeat regions appear to tolerate sequence variation observed in the CSP repeats to bind multiple sequences. Multiple antibodies bind a single repeat with interactions observed between the antibodies as well as between antibodies and CSP, consistent with findings from studies of CSP from *P. falciparum* and P. berghei.

Strengths:

Comprehensive and high-quality studies that examine the conformational states of CSP repeats alongside biophysical measurements to interrogate the interaction of antibodies with CSP repeats and structural studies to define the interactions.

The diverse approaches all align to provide a consistent narrative: the CSP repeats of *P. vivax* are inherently flexible, and antibodies can bind these flexible sequences by leveraging homotypic interactions and tolerating sequence variation within repeats.

CSP is a prominent antigen for the development of vaccines for malaria, and this work may inform further development of *P. vivax* specific vaccines.

Weaknesses:

While no studies have been reported to date on *P. vivax* CSP, the findings are somewhat confirmative of publications from the same group on *P. falciparum* and P. berghei CSP.

The manuscript contains a lot of detailed information that is not easy to follow without a clear simple figure to aid understanding. More proofreading is recommended to minimize errors, improve the content, and increase the clarity of the paper. Specific comments are below (not exhaustive):

1. Line 64. “nonapeptides … or ….” Should be “nonapeptides … and ….”

2. A lot of acronyms were used without being clearly defined. Such as Line 66 “4-aa” ; Line 96 “MD simulation” “KC” “HC” “KCDR” etc.

3. Line 161. Figure 3E referenced in this paragraph did not seem to correlate with Figure 3E. None of the residues mentioned were mentioned in the figure.

4. Line 171. No Figure 2E is found.

5. Lack of consistency. Line 66 “motifs of Pf and P.berghei CSP …”

6. Please use a consistent number format for all Kd values. Such as 1-decimal place is used in Figure 3A and 2- and 3- decimal places were used in Figure 4A.

7. Line 201. Figure 3E should be Figure 4E.

8. Line 618. “HC residues are shown in green and KC residues are shown in grey” better written as “HC is shown in green and KC is shown in grey”.

9. Line 63. “HC residues are shown in blue and KC residues are shown in grey” better written as “HC is shown in green and KC is shown in grey”.

10. Figure 5C figure legend. The definition of “Green highlight” and “Yellow highlight” seems swapped. Please check accuracy.

11. Line 303. Please use “proline” and “glycine” instead of P and G.

---

## [Author Response]

Essential revisions:Overall, the reviewers agree that this paper is strong. The science is rigorous and well described in the manuscript. One area that needs addressing is a more thorough discussion of the biological insights derived from these studies, which, as you know, is a critical component of articles published in eLife.

We thank the Editor and Reviewers for their positive assessment of our work and comments in the Reviews. To expand on the biological insights derived from these studies, we have added the following sentences to three paragraphs in (the Discussion):

“PvCSP peptides have glycine and proline contents of ~22-33% and ~7-11%, respectively, and thus are predicted to fall within the transition region separating amyloidogenic from elastomeric proteins, the latter of which include elastin domains and insect resilin (Rauscher et al., 2006). Moreover, the elastic modulus of PvCSP peptides [~3-5 cal/(mol Å2)] is commensurate with the elastic modulus of peptides modeled after human elastin [~9 cal/(mol Å2)], which requires extensibility and elasticity for its physiological function (Reichheld et al., 2021). This analogy supports the observation that CSP is inherently elastic and suggests that achieving a high degree of conformational disorder may be an essential aspect of CSP function on the surface of sporozoites.”

“Our data indicate that both mAbs 2F2 and 2E10.E9 are cross-reactive as they bind to different repeat motifs of their respective PvCSP variants with similar affinity and in identical conformations. mAbs 2F2 and 2E10.E9 are, however, not cross-reactive to different strains of Pv (Nardin et al., 1982), due to the significantly different CSP sequences of strains PvCSPvk210 and PvCSPvk247. Although both mAbs 2F2 and 2E10.E9 are inhibitory, only mAb 2F2 was reported to induce aberrations in size and in DNA content of sporozoites, indicating a distinct mechanism of attenuation (Roth et al., 2018), possibly due to the higher affinity to PvCSP and slower dissociation rate of mAb 2F2 compared to mAb 2E10.E9.”

“Both mAbs 2F2 and 2E10.E9 recognize their respective epitopes using germline-encoded and somatically hypermutated residues. The difference in affinities between mAbs 2E10.E9 and 2F2 and their respective antigen might partially stem from the low lumber of somatic hypermutations of mAb 2E10.E9 (1 in HC, 6 in KC) compared to 2F2 (14 in HC, 4 in KC). Specifically, mAbs 2F2 and 2E10.E9 use nine and six germline-encoded aromatic residues, respectively to mediate contacts with their core epitopes. Germline-encoded mAb 2E10.E9 residue H.Trp50 forms extensive van der Waals interactions with N6 and P8 residues, analogous to IGHV3-33 germline-encoded H.Trp52 in several human mAbs, including mAbs MGG4 (Tan et al., 2018), 1210 (Imkeller et al., 2018), 311 (Oyen et al., 2017), and other antibodies induced by sporozoite immunization (Murugan et al., 2020) or RTS,S vaccination (Pholcharee et al., 2021) (Figure 3—figure supplement 2 D-E). As both human and murine mAbs appear to depend on germline-encoded aromatic residues for CSP recognition, it is likely that CSP repeats prime the mammalian immune system to select antibodies from germline genes with already optimally-positioned aromatic residues.”

Additionally, the reviewers have made note of a number of specific issues that must be addressed in your revised submission (see below):Reviewer #1:Plasmodium vivax (Pv), which causes Malaria, is decorated with surface receptors that contain central repeat proteins that antibodies can target to prevent Pv infection. Thus understanding the molecular basis of antibody recognition of the peptides in these repeats regions might help advance our basic science understanding and potentially aid in vaccine design. Here, Kucharska et al., used a battery of techniques to characterize the biophysical properties of two strains of Pv repeat peptides, VK210 and VK247, found in the central repeat regions of the Pv receptor protein, Circumsporozoite protein (CSP). A particular strength of this work is the diversity of biophysical techniques used to study the peptides in the absence and presence of inhibitory antibodies, which enabled the authors to provide a comprehensive picture of the biophysical properties of these peptides. However, lacking from the current account is an articulation of the biological insights derived from these studies.1. In Figure 2a, the authors indexed the 210 peptides using letters (a-d), whereas, in other parts of the manuscript, they indexed the peptides using numbers (1-5). The authors should use consistent indexing.

To avoid any confusion, we have revised the peptide indexing, which now includes only numbers.

2. Can the authors clarify the difference between peptides 210-1 and 210-2? In Figure 1, they appear to have the same sequence. Maybe a typo?

We thank the reviewer for noticing this typo, the figure has now been corrected accordingly.

3. The authors report in Figure 1 CD results for the five 210 peptides. However, in Figure 2, they only show MD simulations for four of the 210 peptides. Can the authors comment on why?

To probe for any potential longer-range effects on secondary structure propensities, we performed MD simulations *in silico* on four longer 210 peptides (27-aa) compared to five shorter (18-aa) peptides studied experimentally by CD.

4. The authors should comment on how the elastic moduli they compute compare with other peptides/proteins. For example, do the values fall within the range of what others have calculated for peptides with known elastic properties? This is important because when discussing the molecular basis of 2F2, the authors describe the peptides as elastic, as if self-evident. Are there any easy experiments that the authors can perform to test the MD prediction that these peptides are elastic?

We thank the Reviewer for this excellent suggestion. As such, we have added the following sentence to the Discussion:

“PvCSP peptides have glycine and proline contents of ~22-33% and ~7-11%, respectively, and thus are predicted to fall within the transition region separating amyloidogenic from elastomeric proteins, the latter of which include elastin domains and insect resilin (Rauscher et al., 2006). Moreover, the elastic modulus of PvCSP peptides [~3-5 cal/(mol Å2)] is commensurate with the elastic modulus of peptides modeled after human elastin [~9 cal/(mol Å2)], which requires extensibility and elasticity for its physiological function (Reichheld et al., 2021).”

5. The authors state that "2F2 Fab binds to all tested peptides containing two repeats with similar affinity, with a slight preference for peptide 210-1 (GDRADGQPAGDRADGQPA; 17.0 nM) 137 over 210-2 (GDRAAGQPAGDRAAGQPA; 37.5 nM) (Figure 3A). "However, based on Figure 3a, it seems like the slight preference is force 210-3, not 210-1. Am I mistaken?

We thank the Reviewer for this comment, and have revised the corresponding description to accurately reflect the data presented: “2F2 Fab binds to all tested peptides containing two repeats with similar affinity, with a slight preference for peptide 210-3 (GDRADGQPAGDRAAGQPA; 13.8 nM) over 210-2 (GDRAAGQPAGDRAAGQPA; 37.5 nM) (Figure 3A), with other peptides displaying intermediate affinities (17.1-19.1 nM). Peptides 210-3 and 210-2 differ in the 5th residue of the first repeat, which is an aspartic acid for peptide 210-3 and an alanine for peptide 210-2 (GDRA(D/A)GQPA).”

6. I know that Figure 3e is already a bit crowded, but the authors should consider annotating the image with the distances associated with the contacts they highlighted (dashed lines).

As suggested by the Reviewer, we added additional labels to Figure 3E (A9, G10, K.Gly91, K.Phe96,…) in order to highlight the interactions discussed in the text.

Reviewer #3:[…]The manuscript contains a lot of detailed information that is not easy to follow without a clear simple figure to aid understanding. More proofreading is recommended to minimize errors, improve the content, and increase the clarity of the paper. Specific comments are below (not exhaustive):

We thank the reviewer for their suggestions. Accordingly, we have proofread the manuscript extensively; specific recommendations have been addressed as follows:

1. Line 64. "nonapeptides … or …." Should be "nonapeptides … and …."

Line 64 has been corrected accordingly.

2. A lot of acronyms were used without being clearly defined. Such as Line 66 "4-aa" ; Line 96 "MD simulation" "KC" "HC" "KCDR" etc.

All of the listed acronyms are now clearly defined:

Line 66: 4-amino acid (aa)

Line 96: Molecular Dynamics (MD)

Line 157: The recognition of PvCSPvk210 peptides by 2F2 is mediated mostly by residues localized in Heavy Chain Complementarity-Determining Regions (HCDRs) 1, 2 and 3, and Kappa Chain Complementarity-Determining Regions (KCDRs) 1 and 3.

3. Line 161. Figure 3E referenced in this paragraph did not seem to correlate with Figure 3E. None of the residues mentioned were mentioned in the figure.

As mentioned above in response to Reviewer 2, more labels have now been added to Figure 3E (A9, G10, K.Gly91, K.Phe96,…) in order to highlight the interactions discussed in the text.

4. Line 171. No Figure 2E is found.

This correction has been made.

5. Lack of consistency. Line 66 "motifs of Pf and P.berghei CSP …"

Line 66 was corrected to: …Pf and P. berghei (Pb) CSP…

6. Please use a consistent number format for all Kd values. Such as 1-decimal place is used in Figure 3A and 2- and 3- decimal places were used in Figure 4A.

K_D_ values have been updated throughout the manuscript to three significant figures.

7. Line 201. Figure 3E should be Figure 4E.

Line 201 has been revised accordingly.

8. Line 618. "HC residues are shown in green and KC residues are shown in grey" better written as "HC is shown in green and KC is shown in grey".

Line 618 has been revised accordingly.

9. Line 63. "HC residues are shown in blue and KC residues are shown in grey" better written as "HC is shown in green and KC is shown in grey".

Line 63 was revised accordingly.

10. Figure 5C figure legend. The definition of "Green highlight" and "Yellow highlight" seems swapped. Please check accuracy.

The Figure 5C caption has revised accordingly.

11. Line 303. Please use "proline" and "glycine" instead of P and G.

The nomenclature on line 303 has been revised accordingly.